# The physical roles of different posterior tissues in zebrafish axis elongation

Georgina A. Stooke-Vaughan [1,9], Sangwoo Kim [1,7,9], Shuo-Ting Yen [2,9], Kevin Son[1], Samhita P. Banavar [3,8], James Giammona [3], David Kimelman [4] & Otger Campàs [1,2,5,6] ✉

Shaping embryonic tissues requires spatiotemporal changes in genetic and signaling activity as well as in tissue mechanics. Studies linking specific molecular perturbations to changes in the tissue physical state remain sparse. Here we study how specific genetic perturbations affecting different posterior tissues during zebrafish body axis elongation change their physical state, the resulting large-scale tissue flows, and posterior elongation. Using a custom analysis software to reveal spatiotemporal variations in tissue fluidity, we show that dorsal tissues are most fluid at the posterior end, rigidify anterior of this region, and become more fluid again yet further anteriorly. In the absence of notochord (*noto* mutants) or when the presomitic mesoderm is substantially reduced (*tbx16* mutants), dorsal tissues elongate normally. Perturbations of posterior-directed morphogenetic flows in dorsal tissues (*vangl2* mutants) strongly affect the speed of elongation, highlighting the essential role of dorsal cell flows in delivering the necessary material to elongate the axis.

A holistic understanding of embryonic morphogenesis requires the integration of genetic and molecular events with cell behavior and cell/tissue mechanics. Only in a few examples, and mostly in monolayered epithelial tissues, has it been possible to connect the tissue mechanics guiding morphogenesis to the underlying molecular control, with perhaps *Drosophila melanogaster* gastrulation or germ-band extension as the most prominent examples[1–4]. Looking to examples of 3D tissue morphogenesis, vertebrate body axis elongation is well-studied from a genetic and signaling perspective[5–9] and, more recently, also from a mechanics viewpoint[10–15] making it an ideal system to relate genetic perturbations to changes in tissue mechanics and the resulting tissue flows and morphological phenotype.

The zebrafish tailbud is an excellent system for 3D studies of tissue morphogenesis because the embryos are transparent and develop rapidly compared to amniote embryos, allowing easy tracking of cell and tissue movements by fluorescence microscopy. In addition, techniques developed to measure mechanics in vivo have been extensively used in this system to understand cell and tissue mechanics[10,16,17]. Zebrafish posterior axis elongation starts by the formation of the tailbud through the fusion of the blastoderm margin cells over the ventral yolk plug at the end of epiboly[18]. After the initial formation of the tailbud, a pattern of cell movements is established with cells converging from lateral tissue to the dorsal medial zone (DMZ) of the tailbud and moving posteriorly, then moving to the ventral tissue of the mesodermal progenitor zone (MPZ), from which they are incorporated into the presomitic mesoderm (PSM) (Fig. 1A). As these events occur, the tailbud moves posteriorly, extending the embryonic body axis[19–21]. Previous work has identified the signaling molecules involved in orchestrating tailbud formation and posterior axis elongation[22–24], as well as key molecular underpinnings of specification of neural and mesodermal cell fates from progenitor cells in the tailbud[8,25–29].

[1]Department of Mechanical Engineering, University of California, Santa Barbara, CA, USA. [2]Cluster of Excellence Physics of Life, TU Dresden, Dresden, Germany. [3]Department of Physics, University of California, Santa Barbara, CA, USA. [4]Department of Biochemistry, University of Washington, Seattle, WA, USA. [5]Max Planck Institute of Molecular Cell Biology and Genetics, Dresden, Germany. [6]Center for Systems Biology Dresden, Dresden, Germany. [7]Present address: Institute of Mechanical Engineering, École Polytechnique Fédérale de Lausanne (EPFL), Lausanne, Switzerland. [8]Present address: Department of Chemical and Biological Engineering, Princeton University, New Jersey, NJ, USA. [9]These authors contributed equally: Georgina A. Stooke-Vaughan, Sangwoo Kim, Shuo-Ting Yen. ✉e-mail: otger.campas@tu-dresden.de

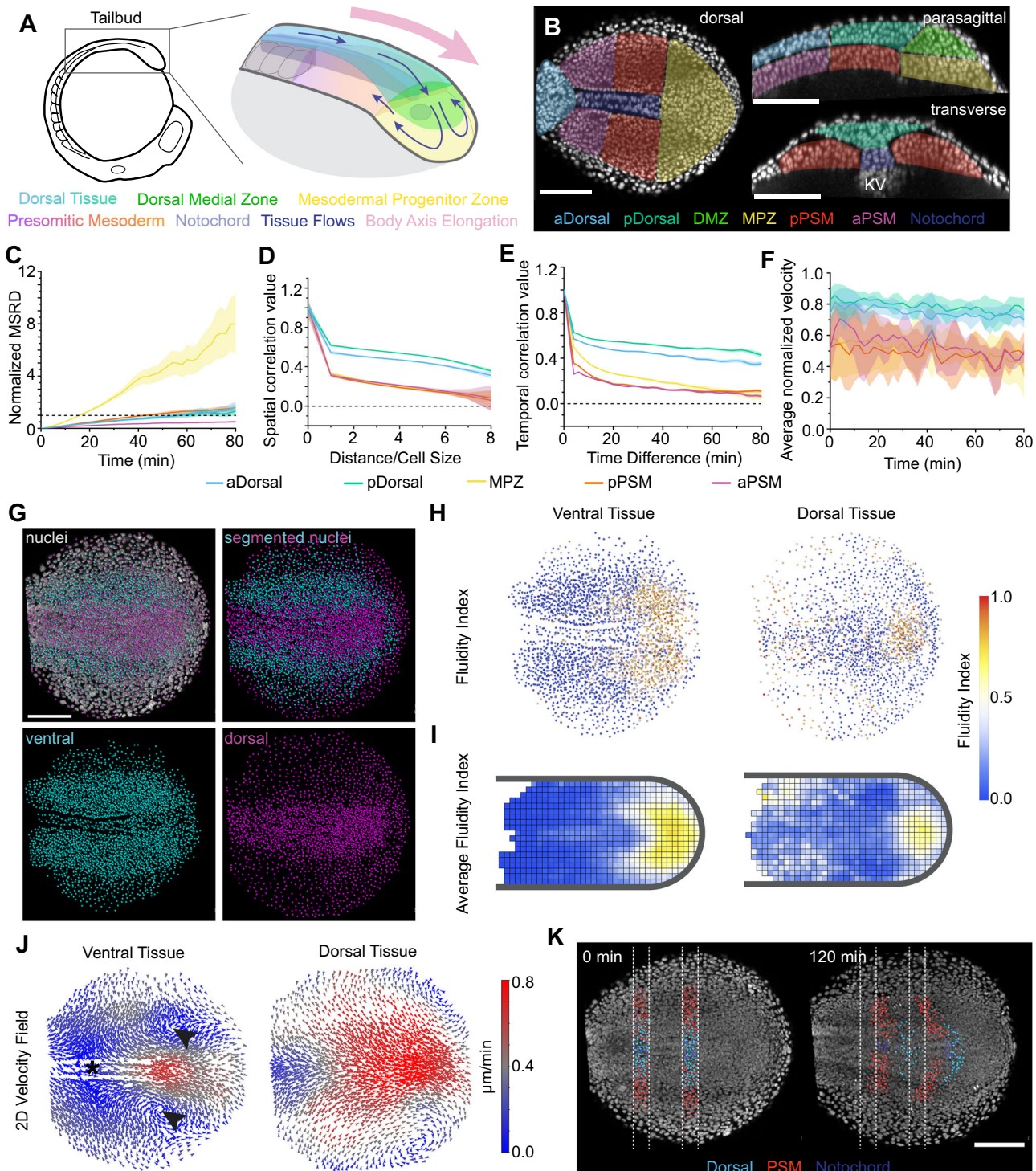

**Fig. 1 | Relative cell movement in wild-type tailbud shows increased fluidity in posterior tissues. A** Schematic of 10 somite stage (SS) zebrafish embryo with the tailbud at top (box), and closeup of major posterior tissues (right; blue and pink arrows are direction of cell motion and axis elongation, respectively). Kupffer's Vesicle (KV) is indicated (gray oval). **B** Representative orthogonal confocal projections through 10 SS tailbud of *Tg(h2afva:h2afva-GFP)* embryo (nuclei; white) showing dorsal, parasagittal and transverse views (N = 5). Colored overlays indicate different tissues. **C–F** Analysis of cell movements in different posterior tissues (shading: SE). Normalized MSRD (**C**), velocity spatial (**D**) and temporal (**E**) autocorrelation, and average normalized velocity (**F**). MPZ: $n_p$ = 3370 nuclear pairs (**C**), $n_t$ = 3831 nuclear tracks (**D–F**) from N = 6 embryos; pPSM: $n_p$ = 2356 (**C**), $n_t$ = 2774 (**D–F**) from N = 6; aPSM: $n_p$ = 1232 (**C**), $n_t$ = 1479 (**D–F**) from N = 5; pDorsal: $n_p$ = 2537(**C**), $n_t$ = 2866 from N = 5; aDorsal: $n_p$ = 2062 (**C**), $n_t$ = 2348 (**D–F**) from N = 5. **G** Example of manual segmentation of the tailbud into dorsal (magenta) and ventral (cyan) tissues. **H** Representative Fluidity Index plot of nuclei from a single wild-type embryo. **I** Averaged fluidity index map of wild-type tailbuds (N = 5). **J** 2D velocity field of averaged cell movements in the tailbud. Arrowheads indicate position of the paired vortices and asterisk indicates position of the stagnation point. **K** Representative example of maximum intensity confocal projections of dorsal views of the tailbud with detected nuclei manually segmented into cells comprising PSM, notochord and dorsal tissue (N = 4). Nuclei were segmented in a transverse section through the tailbud at 0 min, and then tracked over the course of the 120 min timelapse, revealing relative movement between tissues. Scale bars: 100 μm; KV Kupffer's Vesicle; MPZ Mesodermal Progenitor Zone; PSM Pre-Somitic Mesoderm. Source data are provided as a Source Data file.

The mechanics of axis elongation has also been studied, albeit in less detail. In contrast to posterior body axis elongation in amniotes, the posterior elongation of the zebrafish embryonic axis before tail eversion is not driven by proliferation of progenitor cells at the posterior[21,30]. Indeed, body axis extension and tail eversion still proceeds if cell proliferation is inhibited[31–33]. Instead, the combination of spatiotemporal changes in tissue mechanics along the anteroposterior axis and a net posterior flow of cells in dorsal tissues guide axis elongation[34]. Ventral tissues have been shown to undergo a fluid-to-solid transition in their tissue physical state, with the MPZ being in a fluid state and the PSM in a solid-like state[10]. In dorsal tissues, analysis of cell movements showed disordered cell movements at their posterior-most end and more coherent cell movements in the anterior region[20], suggesting changes in tissue mechanics similar to those in ventral tissues. These studies highlight the relevance of spatial changes in tissue fluidity within dorsal and ventral tailbud tissues.

The mechanical interaction between posterior tissues has also been shown to play an important role in axis elongation. Studies in chicken and quail embryos have shown that the PSM provides mechanical support to axis extension[12,35] and that mechanical coupling between the notochord and neural tube, MPZ and PSM is crucial for coordinated axis elongation[35,36]. In zebrafish, posterior elongation after tail eversion from the yolk depends on coordination of anterior and posterior notochord morphogenesis[37], and relative movement of the notochord to PSM is important for shaping somites during maturation[38]. Moreover, before tail eversion, morphogenesis of the neural tube and PSM in zebrafish are coordinated by fibronectin-mediated adhesion between these tissues[39,40]. All these observations suggest that mechanical interactions between posterior tissues may also play an important role in elongation of the zebrafish body axis prior to tail eversion.

Here, we investigate how genetic mutations that specifically affect the PSM, the notochord and the dorsal tissue change the physical state of posterior tissues, as well as the tissue flows shaping the posterior body (morphogenetic flows) and axis elongation before tail eversion. To do so, we first developed an analysis pipeline (and associated automated software) that allows the quantification of multiple physical characteristics of the tissue, including cell movements (Mean Square Relative Displacements, temporal and spatial autocorrelation cellular movements), tissue velocity fields and spatial maps of tissue fluidity. Using this analysis pipeline, we examined the *tbx16* mutant (previously *spadetail*, *spt*), which displays a reduced PSM, the *noto* (*notochord homeobox*, previously *floating head*, *flh*) mutant, which lacks a notochord, and the *vangl2* (previously *trilobite*, *tri*; *strabismus*, *stbm*) mutant, which has defects in convergent extension and radial intercalation movements during gastrulation, resulting in a wide and flat tailbud with very little dorsal medial tissue. We find that the speed of posterior body axis elongation is largely unaffected in the mutants that primarily have defects in ventral tissue morphogenesis. Neither removing the notochord nor disrupting PSM tissues prevents dorsal tissues from extending posteriorly, even if morphogenetic flows in ventral tissues are substantially different. In contrast, reduction of dorsal tissue cellular flow toward the posterior end leads to a reduction in body axis elongation speed. Our results highlight the role of dorsal tissues in supplying the necessary material to build ventral tissues and elongate the axis, as well as the robustness of axis elongation to strong changes in morphogenetic flows.

## Results

### Cell movements and tissue flows in posterior tissues during body axis elongation

We first characterized cell movements and morphogenetic flows in the posterior tissues of wild type embryos from the 10 somite stage (SS) to the 12-14SS, when the tailbud is extending posteriorly along the surface of the yolk and has not yet begun the process of tail eversion. At this stage, neuromesodermal progenitors, which will produce the neural tube and somites of the posterior body, enter the MPZ from the DMZ (Fig. 1A, B). Mesodermal progenitors subsequently move from the MPZ to the PSM, while neural progenitors are incorporated into the developing neural tissue[19,20,29]. We used 3D confocal timelapse microscopy to record nuclear movements in the entire tailbud for 2 h, tracked the cellular trajectories in 3D and validated the accuracy of the tracking procedure (Methods).

To quantify cell movements in different regions of tailbud tissue we developed a custom software—"Cell Movement Analyzer" (CMA; Methods; Supplementary Information) that uses the measured cell trajectories to calculate multiple quantitative measures that characterize 3D cell movements, namely the mean square relative displacement (MSRD), the velocity temporal autocorrelation, the velocity spatial autocorrelation and the average normalized velocity. We used CMA to compare relative cell movements in homogenous cuboidal regions of interest (ROIs from five regions of the tailbud: three ventral regions – MPZ, pPSM, aPSM and two dorsal regions—dorsal tissue dorsal to the aPSM (aDorsal) and dorsal tissue dorsal to the pPSM (pDorsal) (Fig. 1B). MSRD data indicate that cells in the aPSM are caged by their neighbors for at least 80 min, unable to undergo cell rearrangements and exchange positions with their neighbors (Fig. 1C, purple), in agreement with published results showing that the aPSM is in a solid-like state[10]. Cells in the MPZ are uncaged, as indicated by the unbounded MSRD behavior (Fig. 1C, yellow), and the tissue displays a fluid-like behavior, as previously reported[10]. While we previously observed that cells in the pPSM were caged over a 30 min timeframe[10], a small, but finite, level of cell uncaging is observed at timescales above 45 min, indicating a small level of fluidity of the pPSM at these longer timescales (Fig. 1C, orange). Similarly, dorsal tissues show very strongly caged cells for 1 h, but a small level of uncaging after that, indicating a small level of fluidity at timescales longer than 60 min (Fig. 1C, blue and green), in agreement with previous results[20]. Both temporal and spatial velocity autocorrelations, as well as average normalized velocity are higher in dorsal tissues compared to ventral tissues (Fig. 1D–F), indicating that dorsal tissues display more coherent cell movements than ventral tissues, with cells moving together rapidly towards the posterior end of the body, in agreement with previous observations[20].

In tissues with little fibrous extracellular matrix between cells and with relatively compact cells (no highly anisotropic cell shapes), as occurs in the zebrafish embryonic tailbud, relative cell movements can be used to infer the physical state of the tissue[10]. In order to assess variations in relative cell movements across a large, non-homogenous tissue we tracked cells in 3D in the whole tissue (Fig. 1G; Methods) and manually subdivided the tailbud into different regions for analysis: "ventral tissue" consisting of the MPZ, notochord, pPSM and aPSM, and "dorsal tissue" consisting of all tissue dorsal to the PSM and notochord, as well as the DMZ (Fig. 1B). To reveal how tissue fluidity changes in space across the whole tissue, we defined the Fluidity Index (FI), which estimates the probability of local neighbor exchanges and, therefore, the local degree of tissue fluidity, from the local magnitude of relative cell movements (Methods). Specifically, the FI estimates the probability of the tissue being in a fluid state at the location where it is evaluated: FI values close to 1 indicate that the tissue is in a fluid state at a given timescale (30 min unless otherwise stated), whereas values close to 0 indicate solid states. FI can be obtained in 3D at the location of each nucleus (Fig. 1H) or averaged across multiple nuclei to obtain the coarse-grained spatial distribution of fluidity in the tissue (Fig. 1I; Methods). Ventral tissues display the highest value of FI in the MPZ, with the tissue transiting to a solid-like state at the boundary between the MPZ and pPSM, and then behaving like a solid in the aPSM, in agreement with previous direct measurements of tissue mechanics on similar timescales (Fig. 1H)[10,41]. Using landmarks in the tissue to properly align data from different embryos (Methods), we calculated ensemble averages to obtain the stereotypical fluidity map of ventral

tissues (Fig. 1I; N = 5 embryos), which emphasizes the fluid-like MPZ and solid-like PSM. Using the same approach in dorsal tissues, we observed a non-monotonic behavior of the tissue fluidity from posterior to anterior tissues. The FI is highest in a small posterior region where cells enter the MPZ and lowest in the tissue immediately anterior to this region (dorsal of the pPSM), exactly where cells display the fastest, most coherent movements toward the posterior end. However, in more anterior dorsal tissues (dorsal of the aPSM), the FI increases again, albeit not as much as in the posterior-most end of the body. These results reveal spatial inhomogeneities in tissue fluidity in dorsal tissues, with non-monotonic changes along the anteroposterior axis.

To obtain morphogenetic flows from the cellular trajectories, we locally averaged the cell velocities to obtain the velocity field across the tissue (Fig. 1J; Methods). Ventral tissues display high posterior-directed velocities in the MPZ and the posterior end of the notochord, and smaller velocities in more anterior tissues, especially in the aPSM, as previously reported for zebrafish[10,20,34] and avian embryos[12,35,36]. We also observe two counter-rotating vortices flanking the notochord at the transition between MPZ and pPSM (arrowheads, Fig. 1J), as well as a stagnation point with hyperbolic flows surrounding it in the aPSM (asterisk, Fig. 1J), as recently reported[34]. The magnitude of the velocity is lower in the counter-rotating vortices and close to the stagnation point of the hyperbolic flow, as expected theoretically for topological defects of the velocity field[34].

The observed morphogenetic flows generate global deformations in the tissue. Visualization of these tissue scale deformations can be done by photoactivating cells (or photobleaching them) within a region of the tissue[24,30,42]. We instead defined a 30 μm transverse 'slice' through the tailbud tissues after tracking all nuclei, manually classifying tracked nuclei in this slice into dorsal, PSM and notochord tissues, thus allowing the visualization of relative tissue deformations during development from the same tracking dataset (Fig. 1K). Over 2 h of development at 25 °C we see that dorsal tissue moves faster towards the posterior than ventral tissue, and displays a U-shaped deformation indicating that cells at the midline are moving faster than lateral cells, consistent with previous reports of cell movement in dorsal tissues[20,21]. In ventral tissues, we observe that the notochord moves little relative to the aPSM, but there is posterior movement of notochord cells relative to pPSM; correspondingly, we see a greater tissue shear deformation in the pPSM than in the aPSM, suggesting that the shear movements attributed to posterior movement of the notochord and adaxial cells relative to newly formed somites[38,43] may also present at stages prior to tail eversion. These results show that the physical interactions between distinct tissues, such as the notochord and PSM, may be spatially graded, thereby affecting morphogenetic flows differentially across the tissue.

## Posterior MPZ tissues of *tbx16* mutants are substantially less fluid

Having developed and benchmarked an analysis pipeline to characterize cell movements, tissue morphogenetic flows and the spatial variations in tissue fluidity across an entire tissue in wild-type embryos, we then used these techniques to assess how these quantities are affected in mutants with altered tailbud morphology. In particular, we selected three mutants with phenotypes affecting distinct posterior tissues: the *tbx16* mutant, which fails to form PSM; the *noto* mutant, which fails to form notochord; and the *vangl2* mutant, which has aberrant convergent extension movements in dorsal tissues. By investigating one mutant for each of the major tailbud tissues, we aimed to test the requirement of each of these tissues in posterior body axis elongation and tailbud morphogenesis.

We first investigated how cell movements are altered in the *tbx16* (*spadetail, spt*) mutant, which shows an enlarged MPZ and reduced PSM[44] (Fig. 2A), as cells in the *tbx16* MPZ fail to join the PSM[45]. While *tbx16* mutants do not form trunk somites, at 10 SS the

*tbx16* mutant tailbud has begun to form a reduced size PSM due to the compensating activity of Msgn1 and Tbx6l[45–48] (Fig. 2B), which allows *tbx16* mutants to produce a small number of deformed tail somites[44,49]. Despite the reduced size of the PSM, posterior elongation speed of the tailbud around the yolk was unaffected in *tbx16* mutants (Fig. 2C). However, the MSRD in the MPZ of *tbx16* embryos shows a marked decrease compared to wild-type embryos, indicating that the posterior-most tissues in *tbx16* mutants are considerably less fluid (or more viscous) than in wild-type embryos (Fig. 2D). Indeed, direct measurements of tissue mechanics using magnetically-responsive oil microdroplets[10,16] showed an increase in residual stress after 15 min of droplet relaxation, as expected for a less fluid tissue in which stresses take longer to relax (Fig. 2E, F; Methods). These results are consistent with previous studies of movements of mesodermal progenitors showing a slight reduction of global cell movements in dissected posterior tissues[45]. In contrast to wild-type embryos, the temporal autocorrelation of the cell velocity displays a consistent non-monotonic behavior with a characteristic timescale of about 30 min, similar to the periodicity of the segmentation clock (Fig. 2G). The origin of this behavior is unclear, but non-monotonicity in temporal correlations could arise from oscillatory behavior of cell movements, as recently reported in posterior tissues of wild type embryos[50]. While our autocorrelation measurements only suggest the presence of oscillatory cell movements in *tbx16* mutants, this could be a consequence of the more rigid MPZ in *tbx16* mutants amplifying such oscillations. These results reveal that *tbx16* mutants display a less fluid (more viscous) MPZ, potentially hindering the ability of cells in this region from joining the PSM.

In order to reveal potential spatial variations in tissue fluidity, we obtained the FI maps. As in wild-type embryos, the posterior-most tissues display a more fluid-like region, with anterior tissues being solid-like (Fig. 2H cf. Fig. 1H, I). The posterior fluid-like region is larger compared to wild-type, consistent with the morphological phenotype of *tbx16* mutant embryos, which show a characteristic posterior bulging of the tissue (Figs. 1B, 2B). However, the fluidity of these posterior-most tissues (Fig. 2H), including the MPZ in ventral tissues, is strongly reduced compared to wild-type embryos, confirming our MSRD results (Fig. 2D) and direct mechanical measurements (Fig. 2E, F).

Since changes in tissue mechanics are likely to alter morphogenetic flows, we analyzed the velocity field in both ventral and dorsal tissues of *tbx16* mutants. The global structure of the velocity field in ventral tissues completely changes in *tbx16* compared to wild-type (Fig. 2I; cf. Fig. 1J). Specifically, all topological defects in the tissue flows, namely the counter-rotating vortices and the anterior hyperbolic flow are lost, and no retrograde (from MPZ to PSM) flows are observed, with only posterior-directed velocities observed in the tissues (Fig. 2I). In contrast, the velocity field in dorsal tissues is similar to wild-type embryos with respect to direction of flow, although speed of cell movement is slightly reduced. To understand whether a complete loss of PSM would change these observations, we analyzed embryos that completely lack PSM by injecting *msgn1* morpholino (*msgn1* MO) in *tbx16* mutants, which completely recapitulates the phenotype observed in double *tbx16;msgn1* mutants[46,47]. Our results indicate that relative cell movements and velocity fields were the same in the *tbx16* mutants as in embryos completely lacking PSM (Supplementary Fig. 1), confirming our results in *tbx16* mutants. The observed changes in morphogenetic flows translate to changes in tissue deformations. Tracking of cells from transverse slices through the tailbud showed that both ventral and dorsal tissues move posteriorly at similar velocities in *tbx16* mutants, with little to no relative movements between them, indicating minimal shear between these tissues (Fig. 2J). This is in contrast to wild-type embryos, which displayed considerable relative movement between dorsal and ventral tissues (Fig. 1K). These results indicate that both global morphogenetic flows and the relative movements between tissues are strongly affected in *tbx16* mutants.

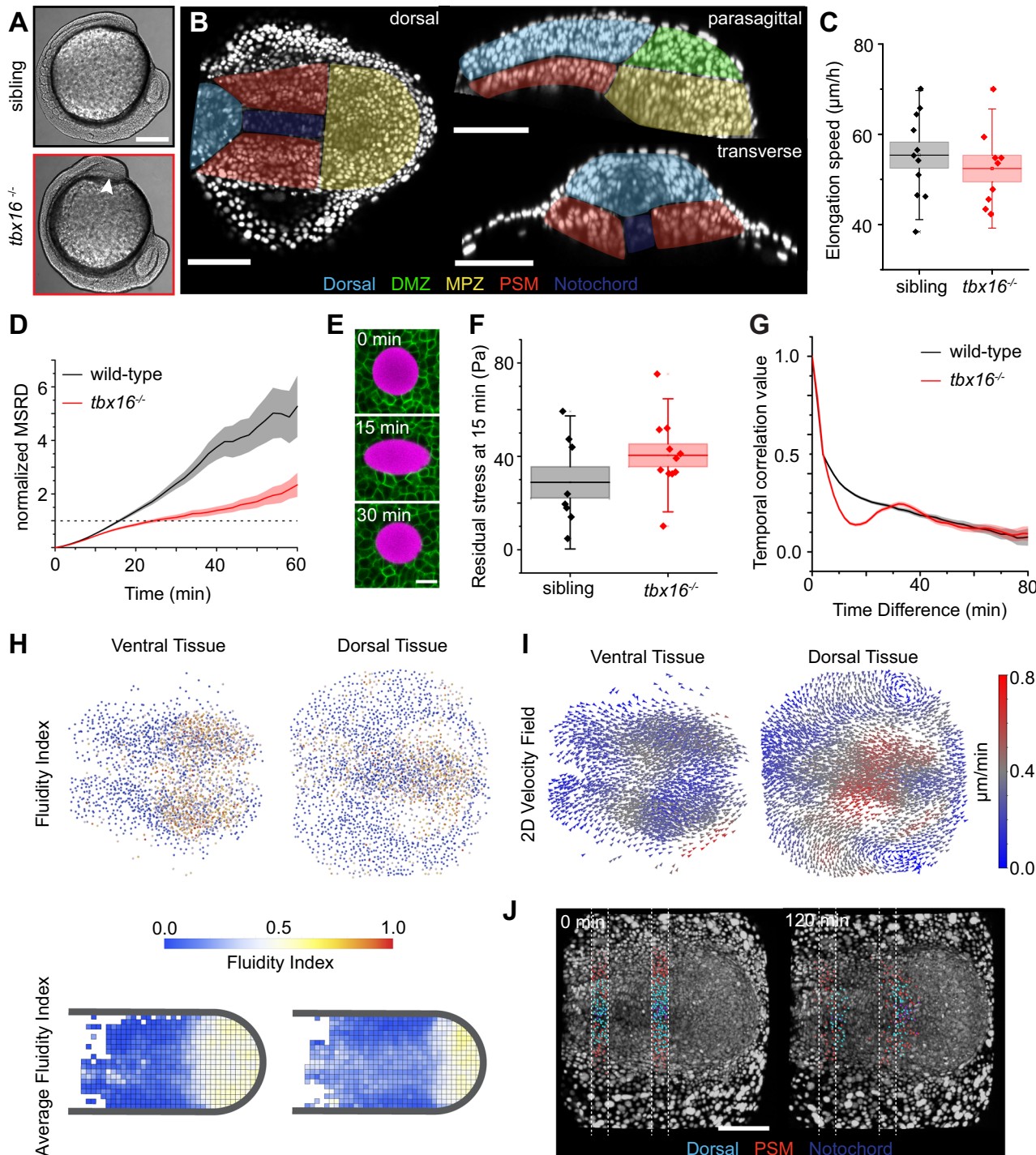

**Fig. 2 | *tbx16* mutants have reduced fluidity in an enlarged MPZ. A** Transmitted light lateral view of 10 SS embryos shows *tbx16* mutants lack somites and have an enlarged DMZ/MPZ (arrowhead). **B** Orthogonal 10 μm confocal projections through 10 SS *tbx16* mutant tailbud with nuclear label (white) from H2B-RFP mRNA injection. Dorsal tissue and MPZ are enlarged relative to wild-type, and the PSM is severely reduced (cf. wild-type in Fig. 1B). Representative image from 5 *tbx16⁻/⁻* embryos analyzed. **C** Posterior body elongation speed is no different in *tbx16* mutants compared to siblings (N = 11 embryos for sibling; N = 9 embryos for *tbx16⁻/⁻*; center line: mean; box: SE; whiskers: SD). **D** Normalized MSRD in the *tbx16* mutant (For wild-type, $n_p$ = 3370 from N = 6, and for *tbx16⁻/⁻*, $n_p$ = 1716 from N = 6; shading: SE). **E** Ferrofluid oil droplet (magenta) embedded in *tbx16⁻/⁻* MPZ injected with mem-GFP mRNA to label cell membranes (green). Panels show droplet before (0 min), at maximum deformation (15 min) and after a subsequent 15 min relaxation (30 min); the residual deformation after 15 min is used to calculate residual stress. Representative image from 11 *tbx16⁻/⁻* embryos analyzed. **F** Residual stress in *tbx16* mutant MPZ compared to siblings (N = 8 embryos for sibling and N = 11 embryos for *tbx16⁻/⁻*; center line: mean; box: SE; whiskers: SD). **G** Velocity temporal correlation in *tbx16* mutants (For wild-type, $n_t$ = 3831 from N = 6, and for *tbx16⁻/⁻*, $n_t$ = 1925 from N = 6; shading: SE). **H** Fluidity index of *tbx16* mutant tailbuds (average fluidity index map from N = 5 embryos). **I** 2D velocity field in a *tbx16* mutant. **J** Maximum intensity projection of a dorsal view of a *tbx16* mutant tailbud with manually segmented and tracked nuclei showing that posterior movement of dorsal tissue relative to ventral tissue is reduced (cf. wild-type in Fig. 1K). Representative image from 3 *tbx16⁻/⁻* embryos analyzed. Scale bar: 400 μm (**A**), 100 μm (**B, J**), 10 μm (**E**). Source data are provided as a Source Data file.

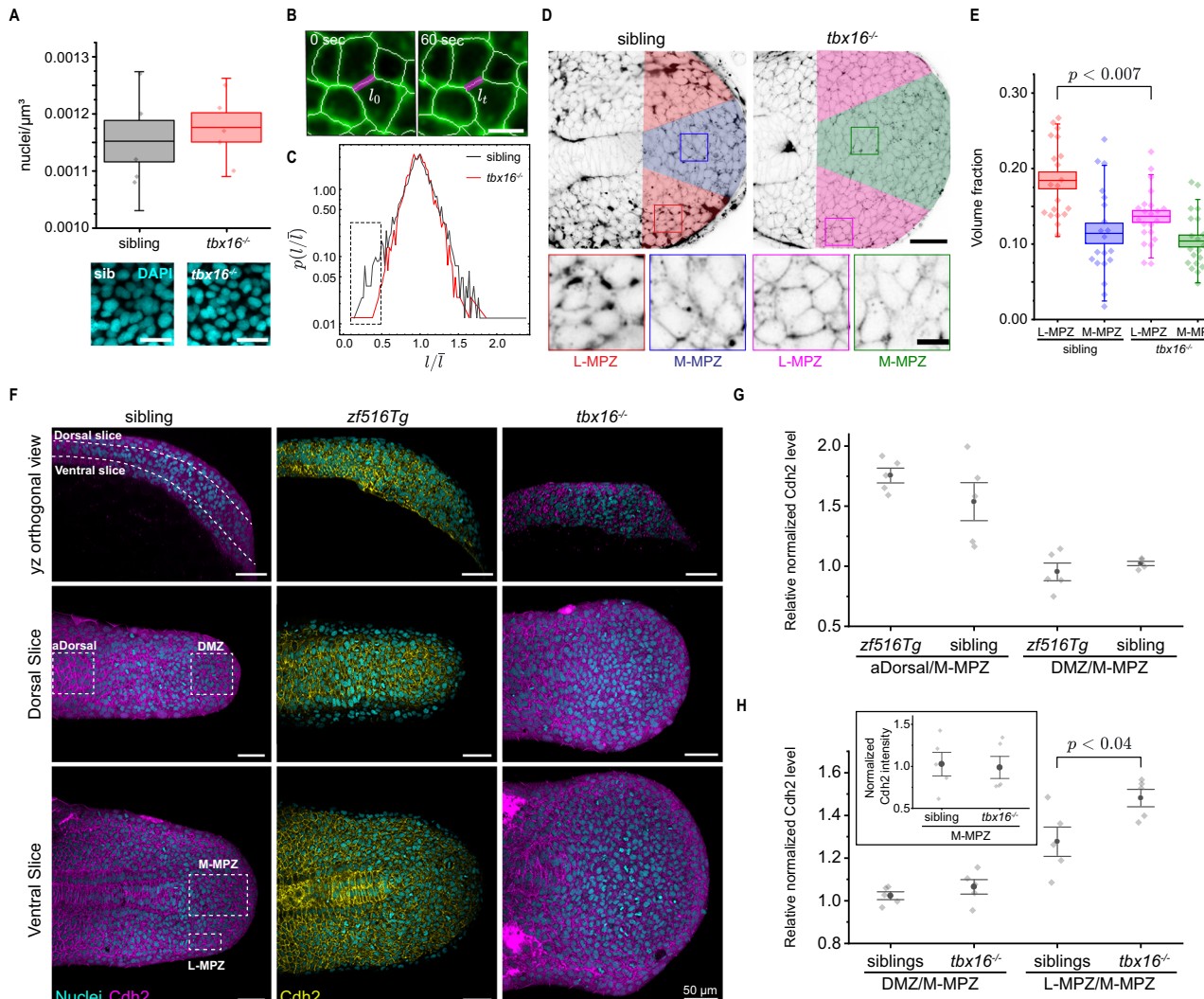

**Fig. 3 | The reduced fluidity of *tbx16* mutant MPZ is due to a decrease in extracellular spaces and junctional fluctuations. A** Measured nuclear density from DAPI stains (insets) (N = 5 embryos for both conditions; center line: mean; box: SE; whiskers: SD). **B** Segmented location (white) of cell membranes (green). Junction length ($l_t$) is tracked over time (magenta). **C** Distribution of normalized junctional fluctuations in *tbx16* mutant (red, n = 4058; N = 4 embryos) and sibling (black, n = 4121; N = 5 embryos) embryos. Dashed box highlights tail of distribution. **D** Representative confocal section showing extracellular spaces (black, inverted LUT). Closeups show zoomed lateral MPZ (L-PMZ) and medial MPZ (M-MPZ). **E** Volume fraction in the L-MPZ and M-MPZ of *tbx16* mutants compared to sibling embryos (*p* = 0.00689; one-way ANOVA with Tukey test for mean comparison) (N = 20 for sib and N = 22 for *tbx16*−/−; center line: mean; box: SE; whiskers: SD). **F** Representative confocal images of dorsal and ventral slices through posterior tissues of transgenic Cdh2 reporter embryos (*zf516Tg*) and Cdh2

immunofluorescence stained (sibling and *tbx16*−/−) embryos (Methods). Slices are defined as dashed lines in orthogonal yz views (parasagittal sections; top left panel). ROIs (white dash boxes; aDorsal. DMZ, M-MPZ, and L-MPZ) were manually defined on these slices for quantification. Magenta: Cdh2 stained with AlexaFluor 594; Yellow: superfolder GFP labeled Cdh2; Cyan: DAPI stained nuclei. **G** Normalized Cdh2 levels in aDorsal and DMZ relative to M-MPZ levels. Quantification for both the transgenic reporter line *zf516Tg* and immunofluorescence staining of wild-type siblings are shown (N = 5 embryos in each group; whiskers: SE; black dot: mean). **H** Normalized Cdh2 levels in the DMZ and L-MPZ relative to values in the M-MPZ for both *tbx16* mutants and siblings (*p* = 0.03671; Mann-Whitney test). Inset: normalized Cdh2 levels in the M-MPZ of *tbx16* homozygous mutants and their wild-type siblings (N = 5 embryos in each group; whiskers: SE; black dot: mean). Scale bars: 20 μm (**A**); 10 μm (**B**); 40 μm (**D**); 10 μm (**D**) inset; 50 μm (**F**). Source data are provided as a Source Data file.

Altogether, our observations reveal that the previously reported reduction in the ability of mesodermal progenitor cells to leave the MPZ and enter the PSM[45] can be attributed to a strong reduction of tissue fluidity in posterior tissues of *tbx16* mutants, leading to an arrest of mesodermal progenitor cells in the MPZ and their accumulation in this region, producing the characteristic *spadetail* phenotype.

**Loss of tissue fluidity in posterior tissues of *tbx16* mutants is due to a Cdh2 dependent reduction of spaces between cells and decreased cell-cell contact length fluctuations**

Since our results indicate a reduction of tissue fluidity in the MPZ of *tbx16* mutant embryos, we studied what physical quantities cells in

*tbx16* mutants are modulating to achieve this change in fluidity. Previous work has shown that tissue fluidity, as well as fluid-to-solid tissue transitions, are controlled by the amount of extracellular spaces between cells and the level of cell-cell contact fluctuations: more spaces between cells and stronger cell-cell contact length fluctuations leading to more fluid tissue states[10,41]. In addition, since cell density could also potentially affect the emergent physical state of the tissue, we first checked if it changed in the MPZ of *tbx16* mutants by measuring the number of cells per unit volume of fixed embryos stained with DAPI (Fig. 3A; Methods). We found no differences between the cell number density in the MPZ of *tbx16* mutants and sibling embryos. By monitoring the lengths of cell-cell contacts over time, we determined

the amplitude of cell-cell contact length fluctuations (Fig. 3B, C). While the average change in the fluctuation amplitude is small, we observed that the tails of the distribution, and especially those fluctuations leading to small cell-cell contact lengths that can cause T1 transitions and fluidize the tissue, were reduced in *tbx16* mutant embryos (Fig. 3C, dashed box). This reduction in large cell-cell contact length fluctuations is likely to contribute to the reduction in tissue fluidity. Finally, we quantified the fraction of extracellular space between cells in *tbx16* mutants and compared it with the fraction in phenotypically wild-type sibling embryos. To do so, we visualized extracellular space in *tbx16* mutant and sibling MPZ by injection of fluorescently labeled dextran (Methods). We found a significant reduction in the volume fraction of extracellular space in the lateral MPZ (L-MPZ) of *tbx16* mutants compared to phenotypically wild-type siblings (Fig. 3D, E). Previous reports suggested increased cell adhesion levels in induced ventral/lateral mesoderm cells from blastula stage *tbx16* mutant embryos[49], which could potentially cause the observed reduction in extracellular spaces. An increase in cell adhesion would also be consistent with recent simulations of tissue dynamics in the tailbud, which predicted that higher adhesion levels lead to reduced spaces between cells, causing a reduction in tissue fluidity[41].

To assess if an increase in cell adhesion in the L-MPZ of *tbx16* mutants could be responsible for the reduction in extracellular spaces, we analyzed Cdh2 (N-cadherin) levels in different regions of both *tbx16* homozygous mutants and their siblings with wild-type phenotype (Fig. 3F–H; Methods). As previously reported[51], Cdh2 levels in some regions of dorsal tissues can be higher than those in the M-MPZ. While no differences in Cdh2 levels were observed between the DMZ and M-MPZ of embryos with wild-type phenotype (both siblings and Cdh2 transgenic reporter line, *zf516Tg*; Methods), our results show that Cdh2 is more abundant in the anterior part of dorsal tissues (aDorsal) compared to M-MPZ levels (Fig. 3F, G; Methods). As for embryos with wild-type phenotype, no differences in Cdh2 levels were observed between the DMZ and M-MPZ in *tbx16* mutants (Fig. 3H, and also inset). However, our results show significantly higher levels of Cdh2 in the L-MPZ of *tbx16* mutants compared to siblings with a wild-type phenotype (Fig. 3H), consistent with the reported lower volume fraction of extracellular spaces in the L-MPZ of *tbx16* mutants (Fig. 3E).

Altogether, these results indicate that the reduction in MPZ fluidity seen in *tbx16* mutant embryos involves a reduction in the largest junctional fluctuations and in extracellular spaces, caused at least in part by increased levels of Cdh2 in *tbx16* mutants. This highlights a role for the Tbx16 transcription factor as an upstream regulator of cell adhesion within the tailbud.

## Absence of notochord disrupts tissue flows but not axis elongation

Since wild-type embryos display shear between the notochord and PSM tissues and it has been reported that the notochord may produce a driving force for axial elongation[36,37], we analyzed the spatial variations in tissue fluidity and morphogenetic flows in *noto* (*notochord homeobox*, previously *floating head*, *flh*) mutants, which do not form a notochord[52]. Although the notochord is absent, no major morphological abnormalities were visible at 10 SS in *noto* mutants, which extended posteriorly at the same speed as wild-type (Fig. 4A, B and also cf. sibling data in Fig. 2C). Without a notochord, the tailbud has dorsal tissue in the space where a notochord would normally be, between the left and right PSM (Fig. 4C cf. Fig. 1B). As development proceeds, the PSM fuses across the midline, resulting in somites that span the midline[52,53]. By defining slices of tissue and tracking its deformation, we observed that the relative posterior movement of dorsal tissues relative to ventral tissues in *noto* mutants is very similar to wild-type, albeit perhaps slightly reduced (Fig. 4D cf. Fig. 1K). This indicates that the PSM deformation observed in wild-type is not solely arising from shear with the notochord, but rather

due to the dynamics of ventral tissues. However, the topological defects characteristic of wild-type morphogenetic flows (counter-rotating vortices and hyperbolic flow) are lost in *noto* mutants. Instead, morphogenetic flows display mostly posterior-directed velocities, with a flow pattern resembling the one in dorsal tissues (Fig. 4E cf. Fig. 1J). The loss of counter-rotating vortices may be because the absence of a notochord leads to a reduced shear on the PSM or that *noto* mutants are characterized by other changes in tissue mechanics[34]. While the size of the fluid tissue region at the posterior-most end of the body axis is reduced in *noto* mutants compared to wild-type, it is still very similar to wild-type (Fig. 4F cf. Fig. 1H, I). Despite *noto* mutants completely missing the normal axial tissue and displaying major defects in morphogenetic flows compared to wild-type, we only see minor defects in body axis elongation prior to tail eversion. Our results indicate that the notochord is not an essential part of body axis elongation at this stage of development.

## Reduction in dorsal posterior-directed cell flow controls rate of axis elongation

Since elongation of dorsal tissues is necessary for body axis elongation, we studied tissue flows in a mutant known for defects in convergent extension, namely the *vangl2* (*VANGL planar cell polarity protein 2*, previously *strabismus*, *stbm*, *trilobite*, *tri*) mutant[54–58]. As a mutant in planar cell polarity, cell movements in *vangl2* are affected during gastrulation resulting in a reduction in both convergent extension and radial intercalation[59,60]. This leads to an embryo with a shortened and flattened body axis at 10 SS (Fig. 5A), with a dorsal tissue reduced to only 1–2 cell layers in depth (Fig. 5B cf. Fig. 1B). Posterior elongation speed is reduced by more than half in *vangl2* mutants compared to siblings (sibling mean elongation speed = 46.5 μm/h, N = 13; mutant mean elongation speed = 16.0 μm/h, N = 9) (Fig. 5C). We found that the average normalized velocity is reduced in *vangl2* mutant dorsal tissue compared to wild-type dorsal tissue (aDorsal and pDorsal ROI's pooled) (Fig. 5D), suggesting that the altered dorsal cell movements in *vangl2* mutants contribute to defects in posterior body axis elongation.

*Vangl2* mutants did not show posterior movement of dorsal tissue relative to ventral tissues and shear in the PSM was strongly reduced despite the presence of the notochord (Fig. 5E cf. Fig. 1K). Since the contact surface area between dorsal and ventral tissues is much larger in *vangl2* mutants, it is possible that a larger mechanical coupling or friction between these tissues causes the lack of relative movements. The morphogenetic flows in ventral tissues of *vangl2* mutants are remarkably similar to wild-type, displaying the counter-rotating vortices as the tissue transits from MPZ to PSM (Fig. 5F cf. Fig. 1J). The *vangl2* dorsal velocity field shows that cells are moving to the midline with less posterior-directed movement than in wild-type embryos (Fig. 5F cf. Fig. 1J). Moreover, the velocity pattern is different, with higher-than-average velocity at the midline instead of at the zone of ingression of cells from dorsal tissue to MPZ (Fig. 5F). Spatial maps of FI in ventral tissues show the same distribution as wild-type, with higher tissue fluidity in the MPZ (Fig. 5G cf. Fig. 1H, I). While the posterior fluid tissue region is shorter and wider in *vangl2* mutants, rescaling the tailbud shape using landmarks shows that it follows the same distribution of fluidity as wild-type embryos (Fig. 5G), indicating that the spatial distribution of tissue fluidity scales with the shape and size of the tissue. These results indicate that posterior-directed cell movements in dorsal tissues are essential for elongating the body axis, as they bring the necessary supply of cells to elongate both dorsal and ventral tissues, and that morphogenetic flows and tissue fluidity in ventral tissues can adapt to changes in tissue shape and size. This suggests that the molecular mechanisms regulating the posterior fluidization of the tissue scale with tissue size.

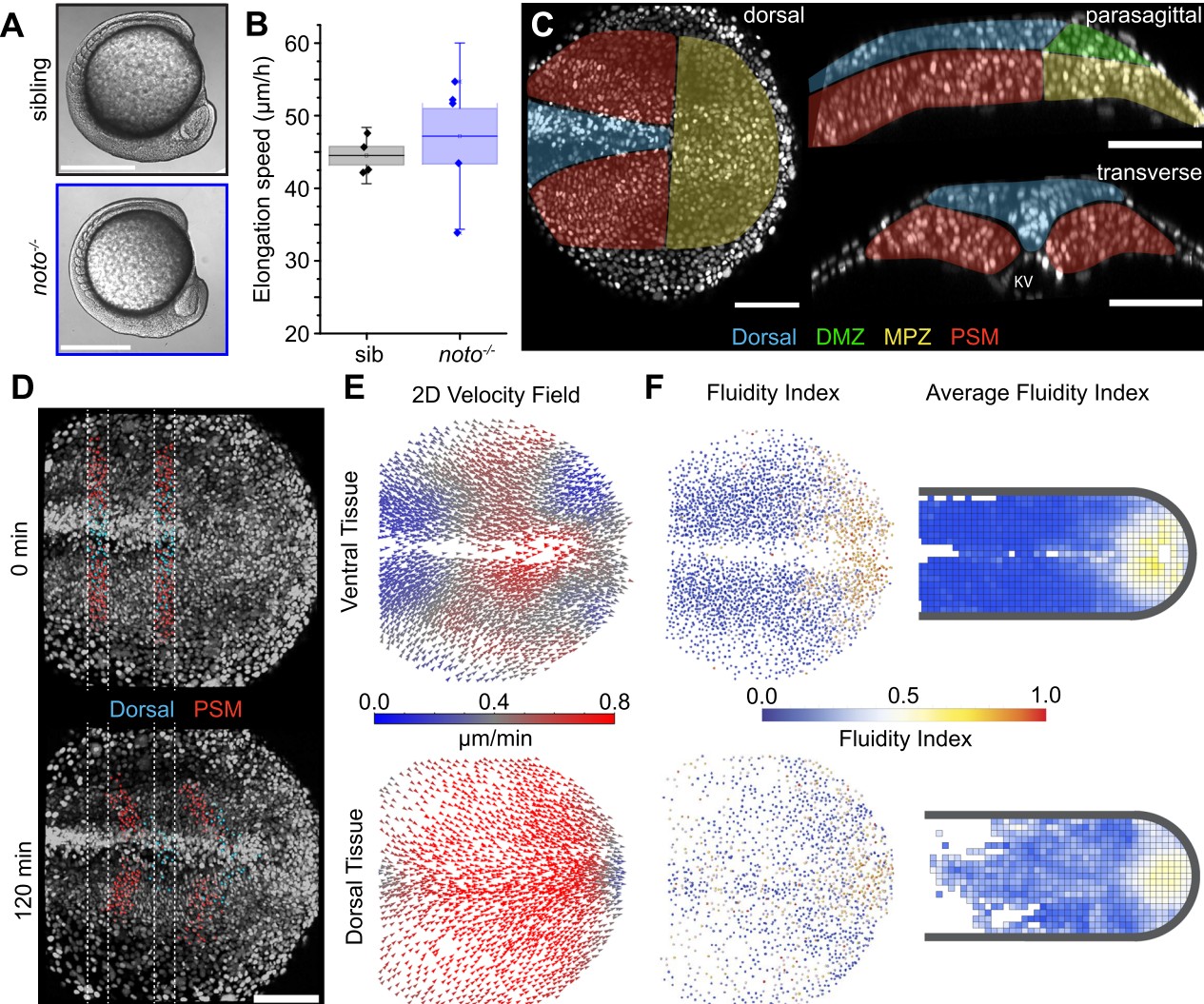

**Fig. 4 | Body axis elongation proceeds in the absence of a notochord.**
**A** Transmitted light lateral view of 10 SS *noto* sibling and mutant embryo – there is no obvious difference in overall embryo shape at this stage. **B** Posterior body elongation speed is no different in *noto* mutants compared to siblings (N = 4 for sib and N = 5 for *noto*−/−; center line: mean; box: SE; whiskers: SD). **C** Orthogonal 10 μm confocal projections of *noto* mutant tailbud injected with H2B-RFP mRNA to label nuclei (white). There is no notochord, and the space usually filled by it is instead occupied by dorsal tissue and PSM. Representative image from 5 *noto*−/− embryos

analyzed. **D** Maximum intensity projection of dorsal view of *noto* mutant tailbud with tracked nuclei showing that tissue shear in the PSM and relative posterior movement of overlying dorsal tissue are comparable to wild type. Representative image from 3 *noto*−/− embryos analyzed. **E** 2D velocity field of *noto* mutant is similar to wild-type apart from the absence of counter-rotating vortices at the MPZ−pPSM boundary. **F** Fluidity index shows no change from the wild-type pattern in *noto*−/− (average fluidity index N = 5 embryos). Scale bar: 400 μm (**A**); 100 μm (**C**, **D**). Source data are provided as a Source Data file.

## Discussion

In this work, we analyzed the spatiotemporal variations in cell movements, morphogenetic flows, and tissue physical state of different posterior tissues during zebrafish body axis elongation, for wild-type embryos as well as in the presence of mutations that affect different posterior tissues. To perform a quantitative analysis of these physical and kinematic quantities, we developed a new software, CMA, to analyze cell movements in different tissue regions, and combined this with a parallel analysis pipeline to obtain the spatiotemporal variations in tissue fluidity and morphogenetic flows across a entire tissue. This pipeline can be used to analyze cell movements in any tissue for which cell trajectories can be measured, as long as there is little fibrous extracellular matrix, cell shapes are not highly anisotropic and cells are not in contact with a boundary. Used in conjunction with molecular methods, these tools can help understand the kinematic and physical changes in tissues resulting from molecular perturbations.

Despite defects in morphogenetic flows and even in the degree of tissue fluidity caused by different genetic perturbations, the regional

fluidization of the posterior-most tissues is always present, indicating that it is a general and necessary feature of posterior axis elongation. Our results reveal that dorsal tissues display a similar fluid-to-solid transition as previously observed in ventral tissues[10], albeit with a partial fluidization of the anterior tissue, which may be associated with the increased cell intercalation rate during convergence extension of this more anterior dorsal tissue. We find that solid-like tissue regions typically display more coherent tissue flows, suggesting that tissue flows may be enhanced and organized by regional tissue rigidification.

We extended our initial observations on wild-type morphogenesis by taking advantage of three mutant lines: one that lacks PSM (*tbx16*), one lacking notochord (*noto*), and one with disordered dorsal cell migration (*vangl2*). Using our new analysis workflow, we could quantitatively explore the changes in the tissue physical state and morphogenetic flows in these mutants, which disrupted each of the key tailbud tissues. In the absence of Tbx16, the mesodermal progenitor zone fails to fully fluidize, with cells never entering the mixing movements associated with the MPZ that are believed to coordinate

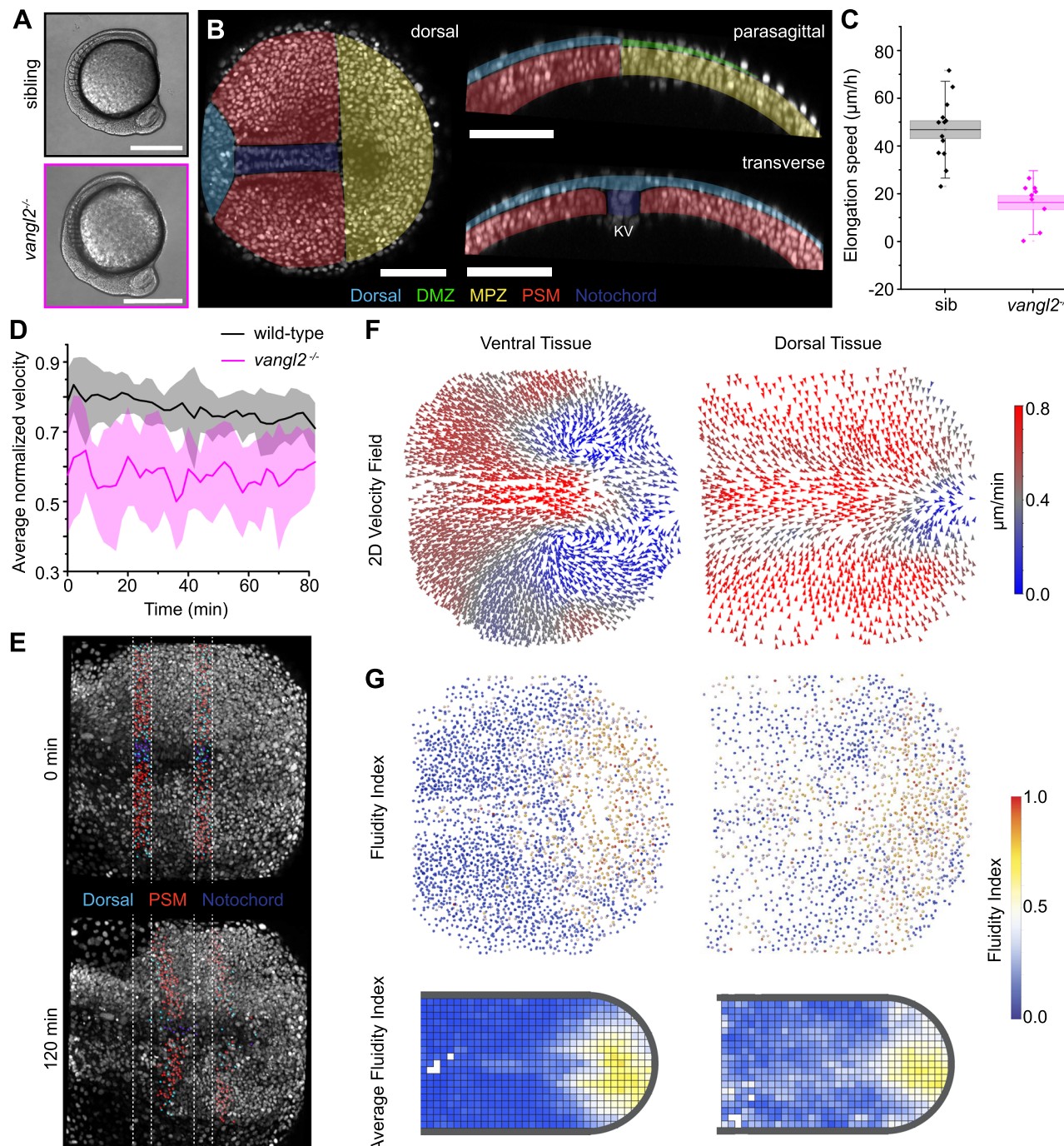

**Fig. 5 | Reduced body axis elongation in *vangl2* mutant is associated with altered dorsal cell movement. A** Transmitted light lateral view of 10SS *vangl2* mutant and sibling embryos—the *vangl2* mutant has a shorter anterior-posterior axis than sibling (9/9 *vangl2*⁻/⁻ and 13/13 siblings analyzed). **B** Orthogonal 10 μm confocal projections of *vangl2* mutant tailbud injected with H2B-RFP mRNA to label nuclei (white) showing the wide and flat shape. Representative image from 5 *vangl2*⁻/⁻ embryos analyzed. **C** Posterior body elongation speed is strongly reduced in *vangl2* mutant embryos compared to sibling (N = 13 for siblings and N = 9 for *vangl2*⁻/⁻; center line: mean; box: SE; whiskers: SD). **D** Average normalized velocity is reduced in *vangl2* mutant dorsal tissue compared to wild type (For wild-type,

$n_t$ = 5214 from N = 10, and for *vangl2*, $n_t$ = 5286 from N = 10; shading SE) **E** Maximum intensity projection of a dorsal view of a *vangl2* mutant tailbud with tracked nuclei showing that there is no relative posterior movement of cells in dorsal tissue with respect to cells in ventral tissue. Representative image from 3 *vangl2*⁻/⁻ embryos analyzed. **F** 2D velocity field of *vangl2* mutant ventral tissues show that the pattern of ventral cell movement is unaffected in the *vangl2* mutant, whereas 2D velocity field of dorsal tissue shows that cells are converging on the midline with less posterior progression. **G** Fluidity Index pattern is unaffected in *vangl2* mutants (average fluidity index N = 5 embryos) compared to wild type. Scale bar: 400 μm (**A**); 100 μm (**B**, **E**). Source data are provided as a Source Data file.

oscillations of the segmentation clock (Fig. 6)[61]. Temporal autocorrelations in cell velocities reveal oscillatory behavior in cell movements in this less fluid MPZ tissue, with a similar periodicity to the segmentation clock, suggesting that changes in tissue fluidity can affect wave propagation in the tissue. Our results indicate that the

failure to completely fluidize the MPZ tissue in *tbx16* mutants arises from a specific Cdh2-dependent reduction in extracellular spaces in the lateral MPZ, indicating that Tbx16 may also be regulating cell adhesion during the somitogenesis stages. The increase in MPZ rigidity in *tbx16* mutants, with a concomitant caging of cells, provides an

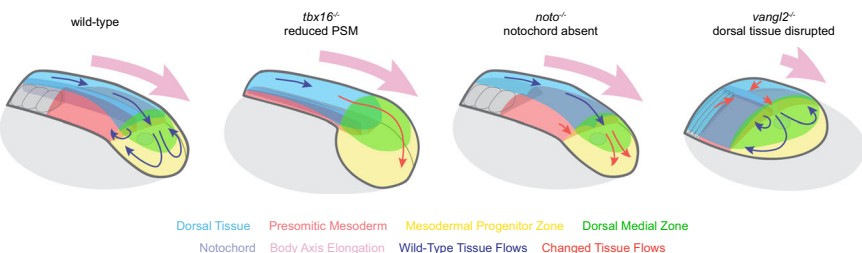

**Fig. 6 | The role of different tissues in posterior body axis elongation.** Diagrams of posterior elongating tissues in wild-type, *tbx16⁻/⁻*, *noto⁻/⁻*, and *vangl2⁻/⁻* mutants, illustrating the high-level tissue flows caused by cell movements. The near-complete failure to differentiate PSM in *tbx16⁻/⁻* does not disrupt posterior body axis elongation, as the enlarged MPZ continues to move posteriorly. The loss of the notochord in *noto⁻/⁻* reduces fluidity in the MPZ and vortices are lost from the MPZ/PSM boundary, but posterior body axis elongation is not disrupted. Disrupted planar polarity movements and dorsal cell movements in *vangl2⁻/⁻* leads to greatly slowed posterior body axis elongation, with ventral tissue movements scaled to the altered proportions of the posterior tissues.

explanation for the increased difficulty of cells to leave the MPZ and form the PSM, which is strongly reduced in *tbx16* mutants. The increased cell caging in *tbx16* mutants fits with previous findings that *tbx16;msgn1* MO knockdown embryos have cells in the MPZ that fail to undergo anterior directed movement or make productive protrusions, despite still producing the same number of protrusions as wild-type cells[45]. More recently, a significant contribution of hyaluronic acid in posterior tissues has been shown to be important for the gradient in extracellular volume fraction[13], so there is potential that the decrease in tissue fluidity seen in *tbx16* mutants may also reflect dysregulation of hyaluronic acid-based extracellular matrix, in addition to the role of Cdh2 reported here.

Despite considerably altering specific tissues, or even in their absence, our observations indicate that axis elongation is robust to such strong perturbations. While wild-type embryos display counter-rotating vortices located at the transition between fluid and solid states in ventral tissues, neither *tbx16* nor *noto* mutants show vortices, instead featuring only posterior directed velocities in ventral tissues (Figs. 2I, 4E; Fig. 6). Recent theoretical work showed that small changes in posterior tissue mechanics can change the structure of morphogenetic flows, with axis elongation proceeding with or without the presence of the vortices[34]. The fact that axis elongation proceeds largely unaffected with completely different structure of morphogenetic flows, as seen in *tbx16* and *noto* mutants, raises the question of whether different cell trajectories, which presumably lead to a different history of the signals perceived by cells, matter to determine cell behavior.

The absence of either notochord or PSM does not seem to affect the ability of dorsal tissues to elongate. Our observations show that dorsal tissues elongate largely unabated in *tbx16* mutants. While this seems at odds with previous studies in *MZoep* mutants indicating that mesoderm is necessary for coordinated movements of the neural plate[62], the extreme phenotype in *MZoep* mutants, lacking essentially all mesoderm and endoderm, makes the direct comparison to our results difficult. Testing how body axis elongation would proceed in the absence of both PSM and notochord would demonstrate that dorsal tissue can elongate on its own, without the need of ventral tissues. The *tbx16;noto* double mutants have previously been created and do extend and form a tail similar to that of the *tbx16⁻/⁻* single mutant, supporting a hypothesis that dorsal tissues have the ability to elongate on their own[63]. However, the double mutants do differentiate anterior notochord in the absence of PSM, so the complete ablation of ventral tissue in zebrafish tailbud is yet to be tested. While before tail eversion the yolk surface provides some level of mechanical support, it is likely that after tail eversion both the notochord and the PSM are necessary for posterior elongation, as recently suggested[35,36,64].

Our investigation of the *vangl2* mutant aimed to disrupt dorsal tissue to understand its contribution to zebrafish body axis elongation. The convergence extension defects in *vangl2* significantly alter dorsal tissue morphology and cell movements, and we found a significant decrease in body axis elongation speed as a result. While defects in the structure of morphogenetic flows in ventral tissues do not significantly alter axis elongation, as seen in *tbx16* and *noto* mutants (Fig. 6), defects in posterior-directed cell flow in dorsal tissues affects axis elongation, as demonstrated by the *vangl2* mutant (Fig. 6). Since the cells necessary to extend ventral tissues come from dorsal tissues, elongation of ventral tissues can only occur if dorsal tissue flows toward the posterior end exist. Therefore, perturbing posterior directed cell movements, as in *vangl2* mutants, reduces movement of material into the tailbud. This means that less tissue is being added to the posterior end of the tailbud and therefore the rate of posterior body axis elongation is reduced. As such, we speculate that any mutant that did not form dorsal tissue would prove lethal at a developmental stage prior to tailbud formation. Beyond defects in dorsal tissue flows, tissue morphology is also altered in *vangl2* mutants (Fig. 5B, c.f. Fig. 1B). However, both tissue flows and fluidity gradients scale with tissue size to the new tailbud proportions (Fig. 5F, G), indicating that the presence and size of the posterior fluid tissue region is important for posterior body elongation. All these observations suggest that convergent-extension in anterior dorsal tissues is the driving force for elongation of dorsal tissues, as it occurs during gastrulation[65] and in agreement with previous observations[19], and that tissue fluidization in the posterior-most tissues is necessary to facilitate tissue remodeling in that region.

Altogether, our results indicate that before tail eversion, posterior elongation is robust to strong perturbations in morphogenetic flows and proceeds even in the absence of ventral tissues.

## Methods

### Zebrafish

Zebrafish (*Danio rerio*) were maintained under standard laboratory conditions[66] and according to protocols approved by the IACUC of UC Santa Barbara. Cdh2 level quantification experiments were performed following all ethical regulations and according to protocols approved by the European Union (EU) directive 2010/63/EU as well as the German Animal Welfare. Lines used were wildtype (AB and TU); *Tg(h2afva:h2afva-GFP)ᵏᶜᵃ⁶* [67]; *TgBAC(cdh2:cdh2-sfGFP-TagRFP,crybb1:ECFP)ᶻᶠ⁵¹⁶* [68]; *tbx16ᵇ¹⁰⁴* [44]; *notoⁿ¹* [52]; *vangl2ᵐ²⁰⁹* [54].

### Injections

For ubiquitous nuclear labeling 40–100 pg of in vitro synthesized 5′-capped *H2B-RFP* mRNA was injected at the 1–2 cell stage. For mosaic experiments to verify nuclear tracking 10–20 pg *H2B-RFP* mRNA was injected into a single cell of a 16–64 cell stage *Tg(h2afva:h2afva-GFP)* embryo. For experiments with the *msgn1* MO (MO1-*msgn1*[46,47]) 40–50 pg *H2B-RFP* mRNA was injected into embryos from a *tbx16⁺/⁻* incross at the 1–4 cell stage, followed by 2 ng *msgn1* MO co-injected with phenol red into the yolk just beneath the blastomeres at 2–8 cell stage; Morpholinos were then taken up into all cells due to cytoplasmic streaming from the yolk and the connections between early

blastomeres in zebrafish embryos. MO-injected embryos were screened at shield stage and any embryos that had not taken up phenol red into cells were discarded.

## Imaging

A Zeiss LSM710 inverted confocal microscope was used for confocal imaging. For full tailbud timelapse to assess nuclear movements, 8–10 SS embryos were mounted in 1% low melting point agarose in a 35 mm MatTek dish and imaged at 25 ˚C using a 25x water immersion objective (LD LCI Plan-Apochromat 25x/0.8 mm Imm Corr DIC M27, Carl Zeiss Inc.). The pinhole was set to a 4 μm section thickness allowing 2 μm z-stack step-size. Stacks through the tailbud were acquired for 2 h, starting at 8–10 somites and ending at 12–14 somites, with a time interval of 2 min. For membrane fluctuations, drop deformation and extracellular space imaging, embryos mounted in a MatTek dish for a dorsal view of the MPZ were imaged using a 40x water immersion objective. Time interval for membrane fluctuation and drop deformation analysis was 5 s and 2.5 s respectively. Extracellular space was imaged by taking z-stacks with pinhole set to 1 AU and optimum z-step size.

For Cdh2 quantification experiments, fixed and stained 8–10 SS embryos (either from *zf516Tg* in-cross or from *tbx16^b104/+* in-cross) were mounted dorsally in 2% (*w/v*) low melting point agarose with Reagent-2 refractive-index matching solution (50% (*w/w*) sucrose, 25% (*w/w*) urea, 10% (*w/w*) triethanolamine, and 0.1% (*v/v*) Triton X-100 in water) in a 35 mm MatTek glass-bottom dish (Part No. P35G-0.170-14-C) and then put on ice with a leveled surface until the agarose jellified. The mounted embryos were imaged on a Zeiss LSM980 confocal microscope using a 40x multi-immersion lens (LD LCI Plan-APOCHROMAT; 420862-9970-799) with glycerin immersion oil (Immersol™ G, Zeiss). The pinhole size and z-stack interval were set to fit Nyquist criteria.

## Visualization of extracellular space and calculation of volume fraction

To visualize extracellular spaces, fluorescently labeled 10 mg/ml dextran-Alexa Fluor 488 MW 10,000 (Invitrogen) in distilled water was injected into either the MPZ or PSM of 8–9 SS *tbx16* mutant and sibling embryos. Embryos were then allowed to recover for at least 30 mins before confocal imaging to acquire z-stacks through the MPZ. Imaris 9.3 (Bitplane) was then used to segment the dextran labeled extracellular space as a surface for a cuboid ROI cropped within the tissue, the Imaris background subtraction algorithm was then used with a threshold of 1.0 and discarding puncta like surfaces with a volume less than 1 μm³. The total volume of surfaces within the ROI was then divided by the total volume of the ROI to calculate the volume fraction.

## Ferrofluid droplets

Ferrofluid droplets were made as described previously[10,16] and fluorescently labeled for confocal imaging using a custom synthesized fluorinated rhodamine dye[69]. 30–40 μm diameter droplets were injected into the MPZ of *tbx16* mutant and sibling embryos at 6–7 SS. Embryos were allowed to recover for 90–120 min before mounting in 1 (*w/v*) % low melting point agarose for a dorsal view of the MPZ. Magnetic stress was applied for 15 min followed by 15 min of relaxation. LabVIEW software was used to fit an ellipse to the droplet during the deformation and relaxation phases as described previously to obtain the droplet interfacial tension[10,16] and the deformation at 15 min was used to calculate residual stress[10].

## Measurement of elongation speed

To measure body axis elongation speed embryos were dechorionated at bud stage and mounted laterally in an imaging dish with 9 cone shaped wells 0.8 mm wide and 0.4 mm deep cast in 2 (*w/v*)% agarose using a 3D-printed mold based on previously published dimensions[70] such that the embryo was held stably in a lateral view without being constricted. Nine embryos were imaged using multi-position timelapse imaging on confocal with 10X objective and 5 min time interval. Elongation speed was calculated from lateral view timelapses with a time interval of 5 min by measuring the distance from the posterior boundary of the 4th somite to the posterior end of the tailbud as described previously[71]. For *tbx16* mutant embryos, which lack somites, embryos were injected with H2B-RFP mRNA at 1–2 cell stage to label cell nuclei. A group of nuclei at a position equivalent to the level of the 4th somite posterior boundary on a stage matched sibling embryo were then followed through the timelapse to measure posterior body elongation.

## Cell movement tracking

Imaging data were first pre-processed using Imaris (version 9.3 Bitplane). Tailbud confocal 3D timelapses were first smoothed using a 1-pixel Gaussian filter. The Normalize Timepoints function was then used to correct for photobleaching, followed by the Attenuation Correction function to correct for z-attenuation. The Free-rotate tool was used if needed to re-align data such that all samples had the same alignment with respect to cartesian coordinates for easy comparison. The fluorescently labeled tailbud nuclei were then segmented using Imaris Spot Detection, and tracked using the Brownian Motion tracking algorithm. Tracking accuracy was determined using mosaic double labeling according to a previously published method[20]; in our case, we found that 67% of tracks were complete, 22% of tracks were split and 9% of tracks were inaccurate mis-tracks that failed to follow a single nuclei (793 double-labeled tracks assessed from 3 embryos).

When using the CMA software, ROIs were cropped in the region of interest before spot detection and tracking. For manual segmentation of dorsal and ventral tissues, spot detection and tracking were first performed on the full tailbud. 30 μm thick transverse slices of data were then created by filtering and duplicating tracks based on "track start position" in one dimension. The Circle Select tool was then used on the 30 μm thick slices to select all nuclei from ventral or dorsal tissue and then duplicated to a new Spots Object. Duplicated nuclei from one tissue where then merged into a single Spots Object slice by slice.

## Cell movements analyzer software

Cell Movement Analyzer (CMA) is a graphical user interface software developed in Python language for quantitative analysis of 3D cell/nucleus movements. Taking cell trajectory information as input data (generally from Imaris software), CMA computes five different quantitative measures; mean squared relative displacement (MSRD), velocity temporal autocorrelation, velocity spatial correlation (absolute), velocity spatial correlation (relative), and averaged normalized speed. For more detail about CMA, see the user guide document for CMA in the supplementary information. The software is also available in the Supplementary Information.

## 2D velocity fields

Nucleus position data from either manually segmented dorsal or ventral tissue or a 30 μm thick ROI was exported from Imaris and used to compute the velocity field using a custom Mathematica script. For each cell trajectory, a B-spline curve was first computed to eliminate high frequency nucleus jiggling motions. Using discretized positions from B-spline curves, velocity values were then calculated by a central difference formula. Velocity values for individual cells were temporally averaged over 12 minutes and were further spatially averaged by taking mean of all velocity values of all cells inside a sphere of 35 μm radius centered at a given cell position. Cell positions and velocity values were then projected on a 2D plane and the velocity field is plotted as an arrow for individual cells. This code is available in the Supplementary Information.

## Fluidity Index

Mean squared relative displacement (MSRD) can be used as an indirect measure for solid/fluid states. Especially, MSRD > 1 implies a separation of initially neighboring cells, and thus a structural rearrangement; therefore, it can be used as a criterion to distinguish solid/fluid states. Here, we defined a fluidity index of a given cell as a probability (between 0 and 1) that MSRD > 1 after a certain amount of time duration, $\Delta t$. $\Delta t$ was typically chosen to be the tissue relaxation time scale and set to 30 min in our analysis.

To calculate Fluidity Index for a single tailbud, positions data from the full dorsal and ventral tailbud were exported from Imaris after manual segmentation of the dorsal and ventral tissues. For each cell at a given time, $t$, eight nearest neighbors were identified based on Euclidean distance and MSRD at $t + \Delta t$ is computed for each pair using a custom MATLAB script. The MATLAB script then computed mean and standard deviation of MSRD values accordingly. Assuming a normal distribution, the fluidity index was computed based on the cumulative probability distribution of mean and standard deviation of MSRD for each cell. The output from the MATLAB script was then plotted using a custom Mathematica script to show Fluidity Index mapped to each cell in the tailbud.

To plot 2D fluidity index maps averaged across multiple tailbuds, full tailbud cell position data was exported from Imaris, and dorsal and ventral regions were separated by a boundary surface, which was interpolated using a custom MATLAB script from boundary points manually identified in Imaris using the Measurement Point tool. 3D cell positions were projected onto a 2D plane of anterior-posterior axis (y-axis) and medial-lateral axis (x-axis) for dorsal tissue and ventral tissue, respectively. The fluidity index for a given location was computed by binning data points in the x-axis and y-axis. To average the fluidity index map over multiple samples, the 2D tail shapes was scaled in terms of tail width (x-axis) and the distance between the posterior end of the MPZ and the notochord (y-axis). After rescaling, the fluidity index maps were further averaged over multiple samples.

## Junctional fluctuation analysis

Cell membranes from 2D confocal timelapses were segmented using the Tissue Analyzer plugin[72] for Fiji[73] with manual corrections applied to minimize segmentation errors. We developed MATLAB pipelines to analyze junctional fluctuations. From a segmented image, individual junctions, individual cells as well as the geometric relation between cell identity and junction identity were first identified. In our previous work, we used a piecewise curve of boundary pixels for junctional length measurements but this method generally overestimates actual junctional length[10]. To minimize discrete pixel effects on junctional length and cell area, the MATLAB pipeline was extended to fit an individual junction with B-Spline curve with the maximum spline degree. From spline curves, junctional length as well as cell area were calculated accordingly. Junctional length depends on the tension fluctuations as well as cross-sectional cell area change. As we were interested in the length fluctuations by the tension fluctuations only, we only sampled junctional length where adjacent cell area varies less than 30% over the entire duration of analyzed images. The exclusion of edges with large adjacent cell area variations was not implemented in our previous work[10] and this step makes our analysis more robust, focusing on quantifying the effect of tension fluctuations on junctional length fluctuations. Individual junctional lengths were normalized and a probability distribution of normalized junction length was computed for each genotype.

## Density measurement

From confocal 3D stacks of fixed MPZ stained with DAPI to label nuclei, nucleus positions were first identified within a box cropped within a homogenous region of the tissue of 80 μm × 80 μm × 40 μm using Spot Detection in Imaris. A custom MATLAB script was then used to calculate density. The script randomly places a small cubic box with a side length L within the cropped region to compute local density. 10,000 random placements of the box were executed and mean density value for a given side length was computed from local density values. The side length was increased from 0.2 μm to 30 μm with an increment of 0.2 μm to test length scale effects on density values. We found that the density value approaches a constant value in the limit of the large box size so a box size of 15 μm was used as an approximation for asymptotic as the density value for a given sample. For each embryo, the density value was computed using the same protocol.

## Tissue clearing and whole mount immunofluorescence staining

To quantify Cdh2 levels in different posterior tissues, we first performed tissue clearing based on the modified CUBIC method with enhanced fluorescence preservation[74–76]. This method maximized light penetrance into deep tissues[77] and minimized optical artifacts. We further modified the method to incorporate (immuno-)staining in between de-lipidation and refractive index matching steps to prevent the staining being lost during de-lipidation. Briefly, 8–10 SS embryos (either from *zf516Tg* in-cross or from *tbx16*[b104/+] in-cross) were fixed with 4%(w/v) paraformaldehyde in PBS for around 5 h at room temperature. After fixation, the embryos were washed with PBS and then cleared with Reagent 1A (10% (w/w) Triton X-100; 5%(w/w) N,N,N',N'-tetrakis(2-Hydroxypropyl) ethylenediamine; 10%(w/w) urea; 25 mM NaCl) de-lipidation solution (50% Reagent 1A in PBS for more than 6 h and then 100% Reagent 1A overnight at room temperature). After de-lipidation, embryos were intensively washed with PBS and then incubated in blocking buffer (5% (v/v) normal goat serum; 0.5%(w/v) Triton X-100 in PBS) at room temperature for more than an hour. Subsequently, embryos requiring Cdh2 immunofluorescent staining were incubated in blocking buffer with primary Cdh2 antibody (1:100 dilution; GeneTex, GTX125962) at 4 °C for over 3 nights. After primary antibody incubation, the embryos were washed several times with PBS-Tx (0.5%(w/v) Triton X-100) at room temperature. Once complete, embryos were incubated in blocking buffer with goat anti-rabbit Alexa Fluor™ Plus 594 secondary antibody (1: 200 dilution; Invitrogen, Cat# A32740) and DAPI (1:1000 dilution) at 4 °C overnight. Then, the embryos were washed several times with PBS-Tx. For *zf516Tg* embryos, only DAPI staining (1:1000 dilution in the blocking buffer) was applied after the blocking step.

After staining was complete, embryos were post fixed with 4% (w/v) paraformaldehyde in PBS for 20 min at room temperature, washed twice with PBS and then exchanged to Reagent-2 (refractive index matching solution; 50% (w/w) sucrose, 25% (w/w) urea, 10% (w/w) triethanolamine, and 0.1% (v/v) Triton X-100 in water) gradually (50% Reagent-2 in PBS for more than 1 hour and then 100% Reagent-2). Finally, the embryos were stored in Reagent-2 at 4 °C before mounting and imaging.

## Quantification of Cdh2 levels

Image stacks for quantification of Cdh2 levels were processed using Fiji[73]. Every image stack of the dorsally mounted embryos was first "resliced" from the "left" to obtain the yz orthogonal view of the stack. The dorsal slice and the ventral slice lines were defined manually on the central section (determined by the morphology of notochord) of the yz orthogonal stack (Fig. 3F). We then used the stack tool, "dynamic reslice", to generate the dorsal and ventral slices of each channel. On these slices, 20 cells in each region of interest (aDorsal, DMZ, M-MPZ, and L-MPZ) were sampled for quantification. Nuclei without condensed chromosomes were first selected for each region and then their cell surface contours were manually defined. The fluorescent Cdh2 light intensity on the cell surface contour of each cell was normalized by its nuclear fluorescence intensity in the same cell. The average normalized intensity of the 20 cells in each region of interest was then compared to that of the M-MPZ, resulting in the relative value of the normalized Cdh2 level.

## Statistics and reproducibility

No statistical methods were used to predetermine sample sizes, but our sample sizes are similar to those reported in previous publications[10,11,16]. Ferrofluid oil droplet deformations were excluded from analysis if the droplet was not deformed above the yield stress of the tissue[10]. Image stacks for Cdh2 level quantification were excluded when the mounted embryo was tilt or not centered. Otherwise, no samples were excluded from the analyses. Analysis of data was done by automated software to ensure blinding and avoid biases in the analysis, except for manual segmentation of tailbud tissues and measurement of body axis elongation speed, where the investigator was not blinded to zebrafish line. No randomization of the data was used.

## Reporting summary

Further information on research design is available in the Nature Portfolio Reporting Summary linked to this article.

## Data availability

Source data are provided with this paper.

## Code availability

The custom-made source code for "Cell Movement Analyzer" (CMA) used in this article is publicly available on GitHub at: https://github.com/campaslab/Cell_movement_analyzer (https://doi.org/10.5281/zenodo.14170064).

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

## Acknowledgements

We thank E. Sletten (University of California, Los Angeles) for sharing custom-made fluorinated Rhodamine dyes. We also thank all past and current members of the Campàs lab for discussions and the UCSB Animal Research Center for support. We also thank the fish facility of

Max Plank Institute of Molecular Cell Biology and Genetics (MPI-CBG) for their technical support and the microscope facility of Physics of Life at TU Dresden for the advice and technical support. GSV was supported by a UCSB Otis Williams Postdoctoral Fellowship. This work was supported by the Eunice Kennedy Shriver National Institute of Child Health and Human Development of the National Institutes of Health (R01HD095797 to OC), and the Deutsche Forschungsgemeinschaft (DFG, German Research Foundation) under Germany's Excellence Strategy—EXC 2068—390729961– Cluster of Excellence Physics of Life of TU Dresden.

## Author contributions

G.S.V. and O.C. designed research; G.S.V. and S.Y. performed all experiments; S.K. and K.S. developed the Cell Movement Analyzer software, with input from S.B., J.G. and O.C.; G.S.V. and S.Y. analyzed the data; D.K. provided reagents and key expertize; G.S.V. and O.C. wrote the paper, with input from D.K., S.K. and S.Y.; O.C. supervised the project.

## Funding

## Competing interests

The authors declare no competing interests.
