## [Transparent Peer Review file · Nature Communications]

The physical roles of different posterior tissues in zebrafish axis elongation

Corresponding Author: Professor Otger Campàs

Version 0:

Reviewer comments:

Reviewer #1

(Remarks to the Author)

This manuscript describes cell flows during zebrafish posterior body elongation. While the authors use some new data analyses, the description of wild-type flows is consistent with those already described in the literature. The major contribution of the study is the analysis of the *tbx16*, *noto* and *vangl2* mutants. While these are well studied mutants, they have never been rigorously examined by a systematic quantification of cells flows as in the present study. The analysis of the *tbx16* mutant is particularly interesting since prior studies might lead one to think that this MPZ region would simply be an expanded of the fluid MPZ region. However, the authors find that it is less fluid. The analyses of the *noto* and *vangl2* mutants is less extensive. However, the findings that the notochord is not required for body elongation at this stage of development, but that convergence extension in the dorsal tissues are required for elongation are solid observations. Overall, the work is of high quality. However, there some issues that need to be addressed before the manuscript would be ready for publication.

Specific comments:

Lines 271-278. The authors suggest that the reduced fluidity in the MPZ of *tbx16* embryos is due to increased cell adhesion. It has been shown that the transition from DMZ to MPZ is accompanied by removal of Cadherin 2 from the cell surface as cells undergo this transition (Das et al., 2017. *Dev Cell* 42:170-180). This study used a *Cdh2* transgene (Revenu et al., 2014 *Development*, 141: 1282-91) which, I believe, the authors have used in a prior paper. The authors should mention these prior results and test whether or not retention of *Cdh2* is apparent in the MPZ of *tbx16* embryos.

Lines 246-247. Manning and Kimelman, 2015. *Dev Biol* 406: 172-85 (ref 43) reported that *tbx16* is required for MPZ cells to transition from a random walk to directed cell movement. This would suggest more cell mixing and a more fluid behavior in *tbx16* mutants. The authors should better explain how the new observations are consistent with the results in reference 43.

Lines 240-244: The logic of these sentences is not clear. If the dorsal and ventral tissues move at similar posterior velocities (the first part of the sentence), then how can there be no differences in their relative movement compared to wild type? How does this suggest reduced shear on the PSM? This sentence is also not consistent with the final sentence of the paragraph as one suggests that relative difference between tissues is maintained while the latter sentence states that the relative movements is strongly affected.

Lines 81-88: This paragraph reviews some prior observation regarding mechanical interaction between posterior tissues. The authors note later mechanical interactions during body elongation but do not note studies of these mechanical interactions prior to zebrafish tailbud eversion which are the most relevant to the present study.

In Figure 2G, the authors report a change in the temporal autocorrelation in the MPZ of *tbx16* mutants. The authors speculate that this non-monotonicity with a timescale of 30 minutes is similar to the period of the segmentation clock. Is this behavior observed on all *tbx16* mutant embryos? It is unclear if this plot contains data from a single embryo or multiple embryos. A recent study of zebrafish tailbud cell flows observed oscillatory cell flows in the PSM of some but not all wild-type embryos (Genuth et al., 2023 *Science Adv*).

In Panel B of each figure, there is panel showing a sectioned view of the embryo labeled "paraxial". This slice view is typically called "parasagittal".

The figure legends should be clearer about the number of embryos analyzed. For example, in figure 1 are panels C-F data from a single embryo or pooled from multiple embryos? If it is a single embryo, how many wild-type embryos were analyzed in this way?

In the main text, it would be useful to briefly define the Fluidity Index. It is essentially a measure of cell neighbor exchange or stability.

Reviewer #2

(Remarks to the Author)

This work studies a critical moment in body axis formation, the formation and extension of the vertebrate tailbud. It takes an approach that combines cell tracking to analyse tissue flow, cell rearrangement and measurements of the physical properties of tissues. By performing these measurements in the contexts of mutants for genes known to result in the absence of specific tissues, it enables the authors to gain insights into how the physical properties of different tissues and their subsequent deformation impact those of neighbouring tissues. From this they draw two major conclusions:

1) The dorsal (neural) tissues undergo a similar fluid-to-solid transition as the PSM.

2) Ventral tissues are dispensable for dorsal tissue extension during these stages of development.

These results are noteworthy, and provide a useful quantitative description of tissue properties and how they are impacted in well studied mutant phenotypes. The paper also presents a new software (CMA) to analyse cell behaviours that will be of interest to others in the field.

I do have some concerns that would need to be addressed prior to publication:

1) The conclusion of a fluid-to-solid transition (stated as a solid-to-fluid transition in the abstract- please check) is occurring in dorsal tissues is based on cell movement analysis and their fluidity index. Direct measurements of the mechanical properties using their bead assay would contribute greatly to this aspect of the paper.

2) At present, there is no causal evidence for the importance of the observed changes in tissue fluidity as a driving mechanism of dorsal (neural) tissue elongation at these stages of development. It seems likely that the convergence and extension movements driving neuroectoderm elongation are playing an important role as gastrulation morphogenesis continues during tailbud formation (Kanki and Ho <https://pubmed.ncbi.nlm.nih.gov/9043069/>). Such an interpretation is consistent with their observations in the *vangl*^{-/-} genetic background. The authors should discuss their work in the context of gastrulation morphogenesis, or else perform further experiments to ascertain whether gastrulation is not the primary cause of the observed developmental defects in the tailbud.

3) Discussion section, beginning line 379. The argument here was hard to follow, and might benefit from a more precise reference to the experimental observations discussed from articles cited. I.e. What exactly is the evidence that is a back-flow is required for PSM formation, and how is this linked to the specific observations of this study? The section also refers the reader to Figure 6, a summary diagram of the work, rather than a specific experiment.

4) The results are clear about the starting stage for each time-lapse dataset (10 somites)- it would be helpful to specify the end stage of their analysis as well.

5) The observations that neuroectoderm elongation continues in the absence of underlying mesoderm tissues should be discussed in the context of known observations about convergence and extension. E.g. <https://pubmed.ncbi.nlm.nih.gov/9043069/>; <https://doi.org/10.1186/1749-8104-9-9>

Reviewer #3

(Remarks to the Author)

In the manuscript by Stooke-Vaughan et al., the authors analyze body axis elongation of zebrafish embryo using various mutants (*tbx16*, *noto* and *vangl2*). They first quantify velocity and fluidity fields of dorsal and ventral tissues in wildtype embryos by using cell tracking data and newly developed software. In *tbx16* mutants, although the volume of PSM (a ventral tissue) is severely reduced, the elongation speed is almost the same as that of wildtype. *noto* mutants that lack notochord also maintain the elongation speed similar to wildtype. The authors find that in *vangl2* mutants, elongation speed becomes less than half of wildtype, due to the reduction of average velocity in dorsal tissue. Based on these experimental data, the authors conclude that dorsal cell flow is essential to driving axis elongation in zebrafish embryo by delivering materials.

This is an interesting work, clarifying a main driver of axis elongation by using mutants with different morphological phenotypes. The manuscript is clearly written. I have a few comments for the authors to further improve the clarity of the manuscript.

[1] The authors show shear deformation between dorsal and ventral tissues in Figs. 1K, 2J and so on. Would it be possible to compute strain rate tensor to characterize local deformation? I thought that subtle differences between mutants can be distinguished by using such local quantity together with global elongation speed.

[2] The authors show that MPZ in *tbx16* mutants is less fluid than that in wildtype. Still, the elongation speed of *tbx16* mutants is almost the same as the wildtype. The authors' previous study (Banavar et al. 2021) showed that the fluidity of MPZ also determines the elongation speed. Could the authors explain why the reduced fluidity in *tbx16* maintains the similar elongation speed to wildtype?

[3] Fig. 5D shows that average normalized velocity in dorsal tissue is reduced in *vangl2* mutants than wildtype. However, their 2D velocity fields (Figs. 1J and 5F) look similar for the most parts of dorsal tissue. I guess that the small region with blue arrows in posterior part in Fig. 5F Dorsal Tissue contributes to the reduction of average normalized velocity. If so, can this small region consistently be observed in several different embryos?

[4] The words "morphogenetic flow" appear several times in the manuscript. However, its definition is missing. I would recommend the authors to provide the definition in the manuscript.

Version 1:

Reviewer comments:

Reviewer #2

(Remarks to the Author)

The authors have addressed all my previous concerns. In my opinion this is now suitable for publication.

(Remarks on code availability)

Reviewer #3

(Remarks to the Author)

The authors addressed all my comments.

(Remarks on code availability)

made.

Reply to Reviewer #1:

This manuscript describes cell flows during zebrafish posterior body elongation. While the authors use some new data analyses, the description of wild-type flows is consistent with those already described in the literature. The major contribution of the study is the analysis of the *tbx16*, *noto* and *vangl2* mutants. While these are well studied mutants, they have never been rigorously examined by a systematic quantification of cells flows as in the present study. The analysis of the *tbx16* mutant is particularly interesting since prior studies might lead one to think that this MPZ region would simply be an expanded of the fluid MPZ region. However, the authors find that it is less fluid. The analyses of the *noto* and *vangl2* mutants is less extensive. However, the findings that the notochord is not required for body elongation at this stage of development, but that convergence extension in the dorsal tissues are required for elongation are solid observations. Overall, the work is of high quality. However, there some issues that need to be addressed before the manuscript would be ready for publication.

We thank this reviewer for the positive review, and also for the comments/questions, which have strengthened the manuscript.

Specific comments:

Lines 271-278. The authors suggest that the reduced fluidity in the MPZ of *tbx16* embryos is due to increased cell adhesion. It has been shown that the transition from DMZ to MPZ is accompanied by removal of Cadherin 2 from the cell surface as cells undergo this transition (Das et al., 2017. *Dev Cell* 42:170-180). This study used a *Cdh2* transgene (Revenu et al., 2014 *Development*, 141: 1282-91) which, I believe, the authors have used in a prior paper. The authors should mention these prior results and test whether or not retention of *Cdh2* is apparent in the MPZ of *tbx16* embryos.

As suggested by this reviewer, we have quantified the endogenous levels of *Cdh2* at cell-cell contacts in dorsal tissues (aDorsal and DMZ) and MPZ (M-MPZ and L-MPZ) of both embryos with wild-type phenotype (siblings) and *tbx16* mutant embryos using immunostaining (with tissue clearing). We cross-checked our results by measuring *Cdh2* levels in the transgenic *Cdh2* line that this reviewer mentioned.

Quantitative fluorescence measurements are notoriously difficult in 3D tissues due to variations in tissue depth and other potential artifacts. To avoid these issues, we measured the values of *Cdh2* fluorescence intensity at cell-cell contacts in each condition and normalized them to the fluorescence intensity of the closest nucleus (DAPI staining). The normalized *Cdh2* intensities were then compared between different regions of each embryo, providing embryo-specific relative values. As mentioned by this reviewer, we observed differences in *Cdh2* levels in dorsal tissues compared to the M-MPZ. In particular, our results show that *Cdh2* is more abundant in the anterior part of dorsal tissues (aDorsal) compared to M-MPZ levels in embryos with wild-type phenotype (siblings; Fig. 3G). However, we observed no differences in relative *Cdh2* levels between the DMZ and M-MPZ both in embryos with wild-type phenotype (siblings) and in *tbx16* mutants. Since the changes in the volume fraction of extracellular spaces between siblings and *tbx16* mutants were only observed in the L-MPZ (Fig. 3E), we analyzed *Cdh2* levels in this region. Our results show significantly larger levels of *Cdh2* in the L-MPZ of *tbx16* mutants compared to siblings with wild-type phenotype, consistent with the reported lower volume fraction of extracellular spaces in the L-MPZ of *tbx16* mutants. We have now included these results in the new version of the manuscript and modified Figure 3 to include these data.

Lines 246-247. Manning and Kimelman, 2015. *Dev Biol* 406: 172-85 (ref 43) reported that *tbx16* is required for MPZ cells to transition from a random walk to directed cell movement. This would suggest

more cell mixing and a more fluid behavior in *tbx16* mutants. The authors should better explain how the new observations are consistent with the results in reference 43.

We agree with the reviewer that Manning and Kimelman, Dev Bio 2015, report a loss of directionality, as well as of persistence, in the movements of mesodermal progenitors in *tbx16* mutants. Our measurements of temporal autocorrelation in cellular movements show a faster decrease in the persistence of cellular movements in *tbx16* mutants, in agreement with Manning and Kimelman, Dev Bio 2015. However, it is difficult to relate tissue fluidity to the absolute cellular movements reported in Manning and Kimelman, Dev Bio 2015, because tissue fluidity is associated with relative cellular movements. Higher fluidity is caused by more frequent rearrangements between cells, and measurements of relative (but not absolute) cellular movements provide a proxy of such rearrangements. This is precisely what our Fluidity Index measures. That said, the measurements by Manning and Kimelman, Dev Bio 2015 indicate that mesodermal progenitors in *tbx16* mutants display slightly lower global movements (distance from origin at 2h, Fig. 3C in Manning and Kimelman, Dev Bio 2015), consistent with our lower MRSD measurements in the MPZ of *tbx16* mutants. So, while it is difficult to compare both studies quantitatively, at least the observations seem qualitatively consistent with each other. We have explained this in the new version of the manuscript.

Lines 240-244: The logic of these sentences is not clear. If the dorsal and ventral tissues move at similar posterior velocities (the first part of the sentence), then how can there be no differences in their relative movement compared to wild type? How does this suggest reduced shear on the PSM? This sentence is also not consistent with the final sentence of the paragraph as one suggests that relative difference between tissues is maintained while the latter sentence states that the relative movements is strongly affected.

Our wording was misleading. We have now rewritten it to make it clear. In contrast to wild type embryos, which display considerable differences in the posterior-directed movements of dorsal and ventral tissues and associated tissue deformations (Fig. 1K), *tbx16* mutants display minimal or no differences in movements between dorsal and ventral tissues (the small PSM observed in Fig. 2J). Tissue shear is directly related to the relative movement between tissues, which is large in wild type and nearly non-existent in *tbx16* mutants. We have clarified these points in the new version of the manuscript.

Lines 81-88: This paragraph reviews some prior observation regarding mechanical interaction between posterior tissues. The authors note later mechanical interactions during body elongation but do not note studies of these mechanical interactions prior to zebrafish tailbud eversion which are the most relevant to the present study.

We thank the reviewer for pointing this out. We have now added a description on mechanical interactions between posterior tissues prior to tail eversion.

In Figure 2G, the authors report a change in the temporal autocorrelation in the MPZ of *tbx16* mutants. The authors speculate that this non-monotonicity with a timescale of 30 minutes is similar to the period of the segmentation clock. Is this behavior observed on all *tbx16* mutant embryos? It is unclear if this plot contains data from a single embryo or multiple embryos. A recent study of zebrafish tailbud cell flows observed oscillatory cell flows in the PSM of some but not all wild-type embryos (Genuth et al., 2023 Science Adv).

We thank the reviewer for highlighting the relation of our finding with this interesting previous work. Our analysis in Fig. 2G is an ensemble average of data from multiple embryos. However, the same behavior is observed if data is taken from a single embryo (see Figure R1 below). This

non-monotonic behavior of the temporal autocorrelation is typically observed in oscillatory behaviors. We do not know if the observed oscillatory cell flows in the PSM of wild type embryos in Genuth et al., Sci Adv, 2023 are related to our observations here, but it may well be possible. We have added a discussion on this point.

Figure R1: Measure temporal autocorrelation in the Progenitor Zone (PZ=MPZ) and in anterior ventral tissues (the partial PSM formed in *tbx16* embryos). Solid full-colored lines are ensemble averages of multiple embryos, whereas lines with transparency are data for individual embryos. The non-monotonic behavior of the temporal autocorrelation in the PZ can be observed both in the ensemble average and also in the traces of most (if not all) individual embryos.

In Panel B of each figure, there is panel showing a sectioned view of the embryo labeled “paraxial”. This slice view is typically called “parasagittal”.

We have now corrected it in all figures of the new version of the manuscript.

The figure legends should be clearer about the number of embryos analyzed. For example, in figure 1 are panels C-F data from a single embryo or pooled from multiple embryos? If it is a single embryo, how many wild-type embryos were analyzed in this way?

We have now explained these in details in the new version of the manuscript.

In the main text, it would be useful to briefly define the Fluidity Index. It is essentially a measure of cell neighbor exchange or stability.

The Fluidity Index is a measure of the local neighbor exchanges. For each cell, we determine its nearest neighbors in 3D and calculate the normalized MSRD of the selected cell with each one of its neighbors. For any fixed time difference (e.g., MSRD(30 min)), we obtain the measured MSRD values with each of the neighbors at that time to construct the distribution of normalized MSRD values at that time. Once the distribution is known, we calculate the Fluidity Index as the probability that the normalized MSRD values are above the threshold value of 1, as this value is known in physics to indicate structural remodeling, or neighbor exchanges, of a particulate material. We have now explained the definition better in the main text.

Reply to Reviewer #2:

This work studies a critical moment in body axis formation, the formation and extension of the vertebrate tailbud. It takes an approach that combines cell tracking to analyse tissue flow, cell rearrangement and measurements of the physical properties of tissues. By performing these measurements in the contexts of mutants for genes known to result in the absence of specific tissues, it enables the authors to gain insights into how the physical properties of different tissues and their subsequent deformation impact those of neighbouring tissues. From this they draw two major conclusions: 1) The dorsal (neural) tissues undergo a similar fluid-to-solid transition as the PSM. 2) Ventral tissues are dispensable for dorsal tissue extension during these stages of development. These results are noteworthy, and provide a useful quantitative description of tissue properties and how they are impacted in well studied mutant phenotypes. The paper also presents a new software (CMA) to analyse cell behaviours that will be of interest to others in the field.

We thank this reviewer for considering our work noteworthy and useful to this field of research.

I do have some concerns that would need to be addressed prior to publication:

1) The conclusion of a fluid-to-solid transition (stated as a solid-to-fluid transition in the abstract- please check) is occurring in dorsal tissues is based on cell movement analysis and their fluidity index. Direct measurements of the mechanical properties using their bead assay would contribute greatly to this aspect of the paper.

We completely agree with this point and we indeed tried to do so. However, dorsal tissues are much thinner than ventral tissues and, as a consequence, injected droplets were consistently extruded from the tissue, precluding any measurements with droplets in dorsal tissues. However, we have previously shown that in posterior zebrafish tissues, the fluid and solid tissue states (as quantitatively measured with magnetic droplets) are directly correlated with the relative cellular movements (Mongera et al., Nature, 2018; Kim et al., Nature Physics, 2021). Moreover, our experiments in wild-type ventral tissues (Fig. 1) show that the fluidity index reports properly the fluid and solid states of the tissues that we previously measured with magnetic droplets, thereby validating this approach.

2) At present, there is no causal evidence for the importance of the observed changes in tissue fluidity as a driving mechanism of dorsal (neural) tissue elongation at these stages of development. It seems likely that the convergence and extension movements driving neuroectoderm elongation are playing an important role as gastrulation morphogenesis continues during tailbud formation (Kanki and Ho <https://pubmed.ncbi.nlm.nih.gov/9043069/>). Such an interpretation is consistent with their observations in the *vangl*^{-/-} genetic background. The authors should discuss their work in the context of gastrulation morphogenesis, or else perform further experiments to ascertain whether gastrulation is not the primary cause of the observed developmental defects in the tailbud.

We completely agree with the reviewer that convergent extension is the driving force for elongation of dorsal tissues. The fluidity changes are important to facilitate tissue remodeling at the posterior end of the tissue. We have added a discussion on this point in the Discussion section of the new version of the manuscript.

3) Discussion section, beginning line 379. The argument here was hard to follow, and might benefit from a more precise reference to the experimental observations discussed from articles cited. I.e. What exactly is the evidence that a back-flow is required for PSM formation, and how is this linked to the specific observations of this study? The section also refers the reader to Figure 6, a summary diagram of the work, rather than a specific experiment.

We agree with the reviewer that our explanation was hard to follow, and it did not help to introduce the term ‘backflow’. We simply meant to say that it does not seem to matter for elongation whether or not the morphogenetic flows display counter-rotating vortices in ventral tissues (as it occurs in wild type), or not (as it happens in *tbx16* and *noto* mutants). We have rewritten this part to make this point clearer.

4) The results are clear about the starting stage for each time-lapse dataset (10 somites)- it would be helpful to specify the end stage of their analysis as well.

We apologize for not having included this. We have added this information in the new version of the manuscript, both in the main text and in the Methods.

5) The observations that neurectoderm elongation continues in the absence of underlying mesoderm tissues should be discussed in the context of known observations about convergence and extension. E.g. <https://pubmed.ncbi.nlm.nih.gov/9043069/>; <https://doi.org/10.1186/1749-8104-9-9>

Our observations indicate that dorsal tissues can elongate in the absence of ventral tissues, with the driving force for elongation in dorsal tissues being convergent extension, consistent with observations reported in Kanki and Ho. On the other hand, comparisons to *MZoep* mutants are difficult because of their severe phenotype. These mutants have a small tail due to an early conversion of the mesendoderm to neural tissue, and a very disorganized neural tissue structure. Consequently, it is difficult to provide a meaningful comparison between *MZoep* mutants and our mutants, which display a rather normal neural tissue architecture and elongating tails. We have now added a discussion on these points in the Discussion section of the new version of the manuscript.

Reply to Reviewer #3:

In the manuscript by Stooke-Vaughan et al., the authors analyze body axis elongation of zebrafish embryo using various mutants (*tbx16*, *noto* and *vangl2*). They first quantify velocity and fluidity fields of dorsal and ventral tissues in wildtype embryos by using cell tracking data and newly developed software. In *tbx16* mutants, although the volume of PSM (a ventral tissue) is severely reduced, the elongation speed is almost the same as that of wildtype. *noto* mutants that lack notochord also maintain the elongation speed similar to wildtype. The authors find that in *vangl2* mutants, elongation speed becomes less than half of wildtype, due to the reduction of average velocity in dorsal tissue. Based on these experimental data, the authors conclude that dorsal cell flow is essential to driving axis elongation in zebrafish embryo by delivering materials.

This is an interesting work, clarifying a main driver of axis elongation by using mutants with different morphological phenotypes. The manuscript is clearly written. I have a few comments for the authors to further improve the clarity of the manuscript.

We thank the reviewer for considering the work interesting, and for the comments, which have clearly helped improve the manuscript.

[1] The authors show shear deformation between dorsal and ventral tissues in Figs. 1K, 2J and so on. Would it be possible to compute strain rate tensor to characterize local deformation? I thought that subtle differences between mutants can be distinguished by using such local quantity together with global elongation speed.

It is indeed possible to obtain the strain rate tensor from the velocity field (Fig. 1J, 2I). However, these velocity fields are obtained in separate cross-sections of ventral and dorsal tissues, rather than on a dorsoventral cross-section spanning both ventral and dorsal tissues. This means that we can obtain the strain rates within each imaging planes (or dorsal and ventral cross-sections), but it is not possible to calculate the strain rate between them because the cross-sections in ventral and dorsal tissues are several cell sizes apart.

That said, the observed differences between wild type and mutants are large enough that our analysis of tissue movements and deformation allows for a qualitative analysis of relative tissue movements.

[2] The authors show that MPZ in *tbx16* mutants is less fluid than that in wildtype. Still, the elongation speed of *tbx16* mutants is almost the same as the wildtype. The authors' previous study (Banavar et al. 2021) showed that the fluidity of MPZ also determines the elongation speed. Could the authors explain why the reduced fluidity in *tbx16* maintains the similar elongation speed to wildtype?

This is a very interesting point. In Banavar et al., 2021, we simulated the extension of ventral tissues alone. In that model, the only contribution from dorsal tissues is a net influx of cells in the MPZ region. However, *tbx16* mutants do not have ventral tissues. Instead, convergent extension in the anterior end of dorsal tissues elongates them posteriorly. The fluid-like dorsal tissues at the posterior end of *tbx16* mutants are simply pushed by the extending dorsal tissue and it does not matter if these are more or less fluid – it is like pushing a ball of cells at the end of the body.

[3] Fig. 5D shows that average normalized velocity in dorsal tissue is reduced in *vangl2* mutants than wildtype. However, their 2D velocity fields (Figs. 1J and 5F) look similar for the most parts of dorsal tissue. I guess that the small region with blue arrows in posterior part in Fig. 5F Dorsal Tissue contributes

to the reduction of average normalized velocity. If so, can this small region consistently be observed in several different embryos?

This is a very good point. This is due to the fact that in *vangl2* mutants, the dorsal tissue flows are more oriented along the mediolateral direction, therefore contributing less to elongation. In wild-type, these flows are more coherent and directed towards the posterior end.

[4] The words “morphogenetic flow” appear several times in the manuscript. However, its definition is missing. I would recommend the authors to provide the definition in the manuscript.

We thank the reviewer for pointing this out. We have now defined morphogenetic flows in the new version of the manuscript.